palaeontology

taxonomy, Triassic, Archosauriformes, body size, Russia

**Author for correspondence:**
Richard J. Butler
e-mail: r.butler.1@bham.ac.uk

# Cranial anatomy and taxonomy of the erythrosuchid archosauriform *'Vjushkovia triplicostata'* Huene, 1960, from the Early Triassic of European Russia

Richard J. Butler[1], Andrey G. Sennikov[2,3], Emma M. Dunne[1], Martin D. Ezcurra[1,4], Brandon P. Hedrick[5,6], Susannah C. R. Maidment[1,7], Luke E. Meade[1], Thomas J. Raven[7,9] and David J. Gower[8]

[1]School of Geography, Earth and Environmental Sciences, University of Birmingham, Edgbaston, Birmingham B15 2TT, UK
[2]Borissiak Paleontological Institute RAS, Profsoyuznaya 123, Moscow 117647, Russia
[3]Institute of Geology and Petroleum Technologies, Kazan Federal University, Kremlyovskaya Street 4, Kazan 420008, Russia
[4]Sección Paleontología de Vertebrados, CONICET—Museo Argentino de Ciencias Naturales 'Bernardino Rivadavia', Ángel Gallardo 470 (C1405DJR), Buenos Aires, Argentina
[5]Department of Earth Sciences, University of Oxford, Oxford OX1 3AN, UK
[6]Department of Cell Biology and Anatomy, School of Medicine, Louisiana State University Health Sciences Center, New Orleans, LA 70112, USA
[7]Department of Earth Sciences, and [8]Department of Life Sciences, The Natural History Museum, London SW7 5BD, UK
[9]School of Environment and Technology, University of Brighton, Brighton BN1 4JG, UK

RJB, 0000-0003-2136-7541; AGS, 0000-0002-1932-0907;
EMD, 0000-0002-4989-5904; MDE, 0000-0002-6000-6450;
BPH, 0000-0003-4446-3405; SCRM, 0000-0002-7741-2500;
TJR, 0000-0002-4349-5635; DJG, 0000-0002-1725-8863

Erythrosuchidae are a globally distributed and important group of apex predators that occupied Early and Middle Triassic terrestrial ecosystems following the Permo-Triassic mass extinction. The stratigraphically oldest known genus of Erythrosuchidae is *Garjainia* Ochev, 1958, which is known from the late Early Triassic (late Olenekian) of European Russia and South Africa. Two species of *Garjainia* have been reported from Russia: the type species, *Garjainia prima* Ochev, 1958, and *'Vjushkovia triplicostata'* von Huene, 1960, which has been referred to *Garjainia* as either congeneric

(*Garjainia triplicostata*) or conspecific (*G. prima*). The holotype of *G. prima* has received relatively extensive study, but little work has been conducted on type or referred material attributed to '*V. triplicostata*'. However, this material includes well-preserved fossils representing all parts of the skeleton and comprises seven individuals. Here, we provide a comprehensive description and review of the cranial anatomy of material attributed to '*V. triplicostata*', and draw comparisons with *G. prima*. We conclude that the two Russian taxa are indeed conspecific, and that minor differences between them result from a combination of preservation or intraspecific variation. Our reassessment therefore provides additional information on the cranial anatomy of *G. prima*. Moreover, we quantify relative head size in erythrosuchids and other early archosauromorphs in an explicit phylogenetic context for the first time. Our results show that erythrosuchids do indeed appear to have disproportionately large skulls, but that this is also true for other early archosauriforms (i.e. proterosuchids), and may reflect the invasion of hypercarnivorous niches by these groups following the Permo-Triassic extinction.

## 1. Introduction

Following the Permo-Triassic mass extinction ([1]; but see [2,3]), terrestrial Early Triassic ecosystems witnessed the beginnings of a major evolutionary radiation of archosauromorph reptiles [4]. Archosauromorpha is a highly diverse clade that includes dinosaurs, birds, pterosaurs and crocodilians, and dominated vertebrate niches in terrestrial ecosystems throughout much of the Mesozoic [5–7]. One of the earliest archosauromorph clades to diversify following the Permo-Triassic mass extinction was the Erythrosuchidae, a group of medium to large-bodied apex predators comprising approximately seven species, known from South Africa, China, India and Russia [6,8–13]. The earliest definitive erythrosuchids are known from the latest Early Triassic, represented by the genus *Garjainia* [14], including the species *Garjainia prima* [14] from Russia and *Garjainia madiba* [11] from South Africa [10,11,14,15].

The type species of *Garjainia* (by original monotypy), *G. prima* [14], is based on a well-preserved partial skeleton comprising a nearly complete skull and fragmentary postcranium, from the Kzyl-Sai II locality, approximately 70 km southeast of the city of Orenburg, in Orenburg Province, Russia (figure 1). Another Russian erythrosuchid genus and species, *Vjushkovia triplicostata* [16], was erected based on well-preserved skeletal remains representing multiple individuals collected at the Rassypnaya locality, west of Orenburg (figures 1 and 2), about 150 km distant from Kzyl-Sai II. The fossils of these two erythrosuchid taxa were both collected from similar stratigraphic horizons, within the Petropavlovskaya Svita of the Yarengian Supergorizont [17], and there have been repeated suggestions that they are synonymous at the generic or even specific level. Tatarinov [18] synonymized both *Garjainia* and *Vjushkovia* with the South African erythrosuchid genus, *Erythrosuchus*, although this synonymy has been rejected by nearly all subsequent authors (e.g. [7,8,10,11,15,19–21]). Some authors have considered *G. prima* and '*V. triplicostata*' to be distinct species, but congeners (e.g. [20,21]), while others have maintained generic as well as specific distinction (e.g. [8,19]). More recently, Gower & Sennikov [15], Gower *et al.* [11], Ezcurra [7] and Ezcurra *et al.* [10] have considered the two taxa conspecific, with '*V. triplicostata*' being a junior synonym of *G. prima*.

The hypodigm of '*V. triplicostata*' includes nearly all parts of the skeleton and represents parts of at least seven individuals. It is the most completely known erythrosuchid, and one of the most completely known archosauromorphs from the Early Triassic. Despite this, it has received little study and its anatomy is poorly documented. Von Huene [16] provided only a brief description, supplemented with highly simplified line drawings of some of the material, as well as cranial and skeletal restorations. Some additional descriptive comments and comparisons were provided by Tatarinov [18]. Subsequent descriptions have focused on specific regions of '*V. triplicostata*': the palate [22], skull and mandible [23], braincase [8,24,25], endocranial cast [26] and tarsus [27]. Gower [9] discussed some aspects of the anatomy of '*V. triplicostata*' in his monograph of *Erythrosuchus africanus*. Ezcurra *et al.* [10] comprehensively redescribed the type material of *G. prima*, but did not describe or figure material of '*V. triplicostata*'. However, this recent work provides an impetus for a more detailed reassessment of '*V. triplicostata*' than has previously been conducted.

To better assess the validity of '*V. triplicostata*', we provide the first full description of the anatomy of cranial specimens attributed to this taxon, and discuss the evidence for its synonymy with *G. prima*. In addition, we quantify, for the first time, one of the features of erythrosuchids often commented

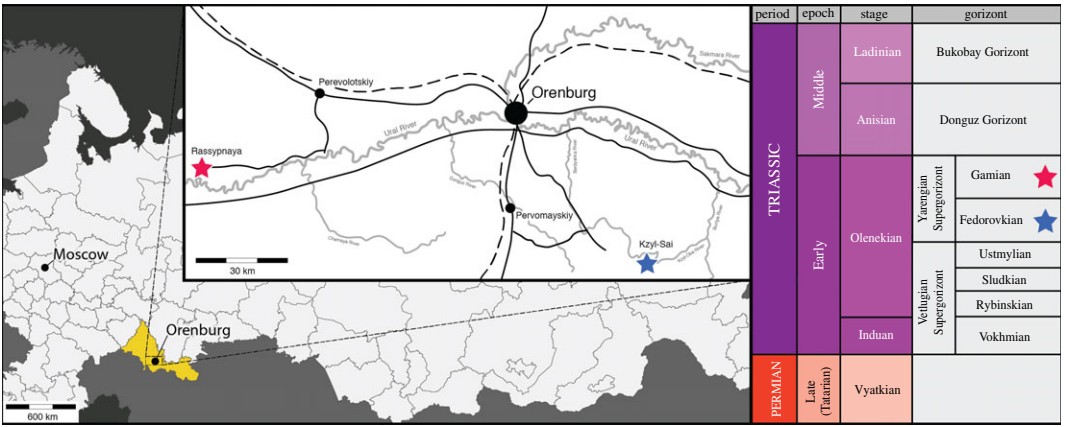

**Figure 1.** Map showing the locations of the Kzyl-Sai (type locality of *G. prima*) and Rassypnaya (type locality of '*V. triplicostata*'). Stratigraphic column shows the major divisions of the late Permian–Middle Triassic of European Russia.

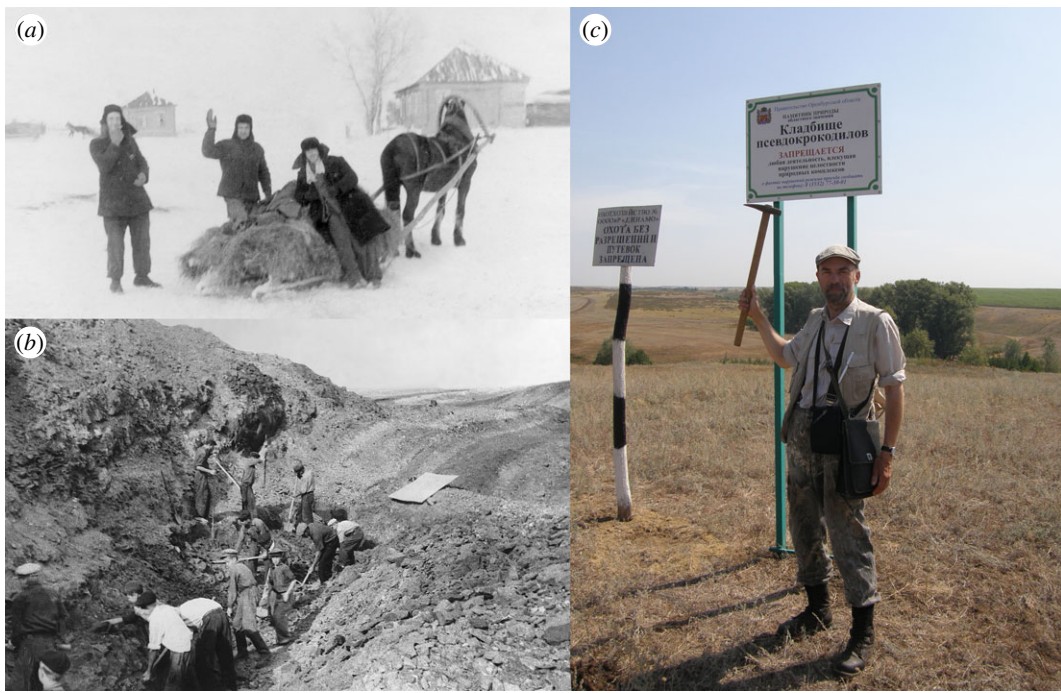

**Figure 2.** (*a*) Rassypnaya locality after excavation in November 1953. From left to right: B.P. Vyushkov, V.A. Garyainov and V.G. Ochev. (*b*) Excavation at Rassypnaya in 1954. (*c*) Rassypnaya locality as geological monument today, with A.G. Sennikov near sign marking the locality. Photographs in (*a,b*) by V.G. Ochev, from archive of M.A. Shishkin.

upon: the relative increase in the size of the skull compared with the postcranial skeleton. We examine the evolution of relative head size among early archosauromorphs, demonstrating that the disproportionately large heads of erythrosuchids are shared with some other early archosauriforms, and hypothesize that they represent an adaptation towards hypercarnivory in these groups.

*Institutional abbreviations*: BP, Evolutionary Studies Institute, University of the Witwatersrand, Johannesburg, South Africa; NHMUK, Natural History Museum, London, UK; NM, National Museum, Bloemfontein, South Africa; PIN, Borissiak Paleontological Institute of the Russian Academy of Sciences, Moscow, Russia.

## 2. Historical background and geological setting

The Rassypnaya locality was found by the geologist V.A. Garyainov from Saratov State University during geological mapping of the Orenburg region in the autumn of 1953 [28]. He passed the details

of the site to I.A. Efremov and B.P. Vyushkov, palaeontologists at the Paleontological Institute of the Academy of Sciences of USSR in Moscow. Vyushkov promptly organized an expedition to excavate the fossils, and invited V.A. Garyainov and V.G. Ochev to participate (the latter was at that time a student at Saratov State University). The excavations started towards the end of November 1953, with the scientists being assisted by local school children. The fossil bones of 'V. triplicostata' were preserved in claystone and were fragile, proving difficult to excavate and prepare. An unexpected storm during the excavation covered the locality and surrounding steppe with more than a metre of snow, meaning that fossils had to be transported to the railway station using horses and carts (figure 2a). In 1954, B.P. Vyushkov organized a second field season to continue the excavation (figure 2b), including the first use of a bulldozer in Russian palaeontological fieldwork [29]. Bones were collected from a minimum of six individuals of 'V. triplicostata', showing a substantial size range. More recently, in 1974, M.G. Minikh, a palaeontologist from Saratov State University, discovered at the same locality and claystone horizon the remains of an additional large individual that he identified as 'V. triplicostata' (M.G. Minikh 2019, personal communication to A.G.S.). Currently, the outcrop and excavation site are not exposed, being covered by landslips and vegetation (A.G.S. 2019, personal observation), but the locality is listed as a geological monument of the Orenburg region (figure 2c).

The type specimen of G. prima and the type material of 'V. triplicostata' come from the Yarengian Supergorizont (figure 1), where the fauna have been subdivided into two tetrapod biochrons corresponding to the Fedorovkian and Gamian gorizonts [17]. The type locality for G. prima, Kzyl-Sai II, occurs within the lower of these two biozones, the Fedorovkian. By contrast, the type locality of 'V. triplicostata', Rassypnaya, occurs with the upper biozone, the Gamian. Thus, there is a potential temporal difference between the two species, with 'V. triplicostata' being stratigraphically younger. Both the Fedorovkian and Gamian gorizonts are dated as latest Early Triassic (late Olenekian) in age, based on the occurrence of the index genus for the Yarengian (the temnospondyl Parotosuchus) in strata containing biostratigraphically informative ammonites and miospores [17].

Associated fauna from the Rassypnaya locality include the temnospondyl Parotosuchus orenburgensis, the archosauriform Chasmatosuchus magnus (='Jaikosuchus magnus'; [7,15,21,30,31]), the non-archosauriform archosauromorph Vritramimosaurus dzerzhinskii [32], and the therocephalian Silphedosuchus orenburgensis [33].

# 3. Methods

Erythrosuchids are often described as possessing disproportionately large heads (e.g. [6,8–10,18,34]), but this trait has never been explored quantitatively. In order to explore relative head size in erythrosuchids in more detail, we collected data on basal skull length (premaxilla to quadrate) and femur length across tetrapods, with a particular focus on early archosauromorphs. Femoral length was chosen as a proxy for overall body size, in line with many previous studies (e.g. [35,36]).

Basal skull length and femur length for 89 species of extant mammals, lepidosaurs and crocodilians, and fossil dinosaurs and pterosaurs were drawn from VanBuren et al. [37]. To these, we added data from the literature and personal observations for 41 species of Early Triassic to earliest Jurassic archosauromorphs, including non-archosaurian archosauromorphs, pseudosuchians, pterosaurs and early dinosaurs (see electronic supplementary material). All data were $\log_{10}$ transformed prior to further analyses. These data included three erythrosuchid species. For E. africanus, NHMUK R3592 includes a partial skull and complete femur [9], and a complete basal skull length was estimated by scaling the partial skull against the complete skull of the referred specimen BP/1/5207 (electronic supplementary material). For 'V. triplicostata', we estimated basal skull length for the largest known and most complete cranial material, the lectotype PIN 951/59, based on comparison with the holotype of G. prima [10]. Basal skull length was estimated at 595 mm. Femur length was based upon PIN 951/61-1, the largest known femur with a length of 243 mm. Although it is uncertain whether PIN 951/59 and PIN 951/61-1 represent the same individual, this nonetheless provides constraints on the possible skull–femur ratio. For Shansisuchus shansisuchus, we used measurements for the largest skull and largest femur described by Young [38], although, as for 'V. triplicostata', it is uncertain whether these represent a single individual.

Quantitative analyses were conducted in the statistical environment R [39]. We first compared basal skull length and femur length using standard major axis regression with 95% CIs using the package lmodel2 [40]. To further explore these results in a phylogenetic context, we created an informal supertree for the 42 Triassic and earliest Jurassic archosauromorphs in the dataset. This supertree was

built following the topologies recovered by Ezcurra [7] and its subsequent iterations (e.g. [4,10,12] for most of the tree, and Langer *et al.* [41] for rhynchosaurs, Nesbitt [5] for loricatan pseudosuchians and Ezcurra [42] for dinosaurs. The supertree possesses polytomies at the base of Tanystropheidae, Rhadinosuchinae and Avemetatarsalia in order to acknowledge unresolved relationships in the above-mentioned published topologies. The supertree was timescaled using the function timePaleoPhy of the package paleotree [43] with the minimum branch length set at 1 Myr. Stratigraphic ages for species were taken from the literature. We examined the correlation between basal skull length and femur length using phylogenetic generalized least squares with the gls function of the package nlme [44]. We then tested for the presence of a phylogenetic signal in the ratio of basal skull length to femur length using the phyloSignal and phyloCorrelogram functions of the package phylosignal [45], setting 999 999 and 9999 replicates for each function, respectively. The lipaMoran function of phylosignal, setting 999 999 replicates and a significant $p$-value < 0.05, was used to identify phylogenetic associations within the data—i.e. groups of species with similar values—and the results were visualized using the barplot.phylo4d function.

# 4. Systematic palaeontology

Diapsida [46] *sensu* Laurin [47]

　　Sauria [48] *sensu* Gauthier, Kluge & Rowe [49]

　　Archosauromorpha [50] *sensu* Dilkes [51]

　　Archosauriformes [49]

　　Erythrosuchidae [52] *sensu* Ezcurra *et al.* [53]

　　*Garjainia prima* [14]

　　*Synonymy*: Vjushkovia triplicostata [16]

　　*Holotype*: PIN 2394/5 (formerly SGU 104/3-43), partial skeleton of a single individual, from the Kzyl-Sai II locality, Orenburg Province, Russia (see [10]).

　　*Referred specimens*: PIN 951/59, a skull roof and braincase (pl. 11, figs 1 and 2 in [16] , fig. 2 in [18]; fig. 8.4C in [15]), lectotype of 'V. triplicostata'; numerous isolated skull and postcranial elements representing paralectotypes of 'V. triplicostata', including parts of nearly the entire skeleton, representing at least six individuals (see appendix A for full list of 'V. triplicostata' paralectotype cranial and mandibular material); and PIN 951/69, mandibular material of a large individual collected by M.G. Minikh in 1974 (see appendix A for details).

　　*Horizon and locality for referred specimens*: Rassypnaya locality, Petropavlovskaya Svita of the Yarengian Supergorizont, Gamian Gorizont (Early Triassic: late Olenekian), 1.5 km northeast of Rassypnoe village, right bank of the Ural River, Ilek district, Orenburg Province, Russia ([15,16,21]; locality 51 of [54]). All referred specimens, including the lectotype and paralectotypes of 'V. triplicostata', are from the same locality.

　　*Remarks*: Synonymy of 'V. triplicostata' with *G. prima* has been suggested previously [10,15] and is supported here. Detailed discussion of the proposed anatomical differences between the two species is provided below.

# 5. Cranial and mandibular anatomical description of 'Vjushkovia triplicostata'

## 5.1. Premaxilla

A pair of premaxillae is present (PIN 951/63; figure 3*a–e*), preserved in articulation. They are mostly complete, although their prenarial processes have mostly broken away. A second fragment of a left premaxilla (PIN 951/116; figure 3*f,g*) is present and represents a larger individual. In PIN 951/63, much of the left postnarial process and the posterior margin of the right postnarial process are missing (figure 3*a,b*). The main body of the premaxilla is subrectangular in lateral view and is at least 2.25 times longer than the distance between the alveolar margin and the ventral border of the external naris. The base of the prenarial process is narrow in lateral view and transversely compressed, and its orientation indicates that the process was posterodorsally directed. The postnarial process is a posterodorsally directed, transversely compressed plate. Several foramina and grooves are present on the external surface of the main body of the premaxilla. Some anteriorly opening foramina are

royalsocietypublishing.org/journal/rsos　　R. Soc. open sci. **6**: 191289

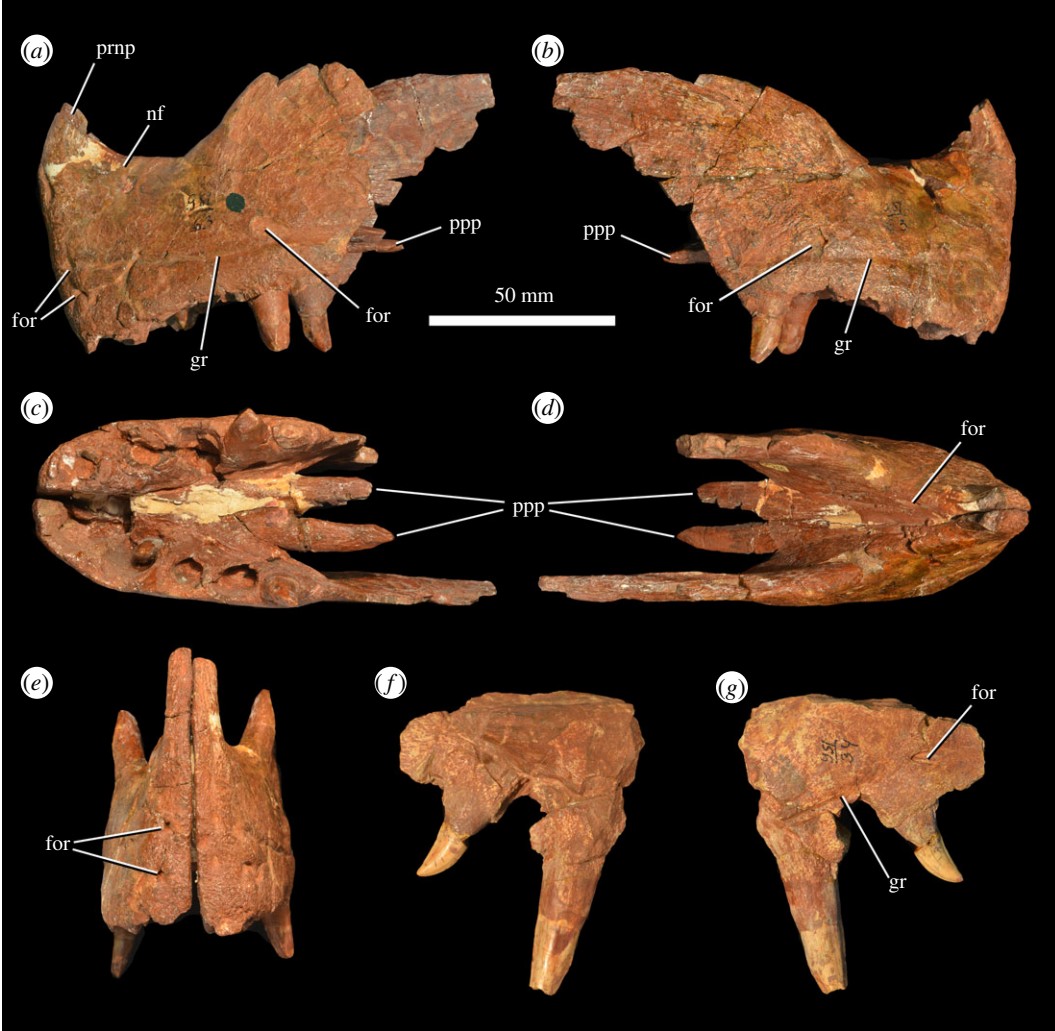

**Figure 3.** Premaxilla of *G. prima* (='*V. triplicostata*'). Articulated left and right premaxillae (PIN 951/63) in left lateral (*a*), right lateral (*b*), ventral (*c*), dorsal (*d*) and anterior (*e*) views. Fragment of left premaxilla (PIN 951/116) in medial (*f*) and lateral (*g*) views. for, foramen; gr, groove; nf, narial fossa; ppp, palatal process of premaxilla; prnp, base of prenarial process.

clustered together at the anteroventral corner of the bone (figure 3*a,e*). A groove extends horizontally across the main body of the lateral surface of the premaxilla, roughly parallel with the tooth row, approximately 19 mm above the alveolar margin anteriorly and approximately 8 mm above the alveolar margin at its posterior end (figure 3*a,b*). The presence of this groove has been considered a synapomorphy of the genus *Garjainia* [7,10,11]. This groove is associated with a number of foramina in '*V. triplicostata*', including a large posteroventrally opening foramen below the base of the postnarial process (figure 3*a,b*). There is a shallow and dorsally restricted narial fossa adjacent to the ventral margin of the external naris (figure 3*a*). The lateral surface of the postnarial process is medially recessed relative to the main body of the bone and the process is transversely thickened at the margin of the naris. The palatal process is preserved on each premaxilla (figure 3*a–d*). The process is directed posteriorly and does not contact its antimere along the midline. The orientation of the process in lateral view suggests that the anterior part of the premaxilla was downturned relative to the maxilla. The palatal process is visible in lateral view posterior to the main body of the bone. By contrast, the palatal process of the premaxilla is proportionally shorter in the holotype of *G. prima* (PIN 2394/5) and is completely obscured by the main body of the bone in lateral view [10]. In dorsal view, there is a foramen present on the medial surface of the premaxilla, immediately ventral to the ventral margin of the external naris (figure 3*d*).

Each premaxilla bears five alveoli (figure 3*c*). Teeth are absent from most of these, although the fourth left tooth and fifth right tooth are preserved *in situ* and are fully erupted. A replacement tooth is also visible in the second alveolus on the right side. The two *in situ* teeth are fused at their bases to

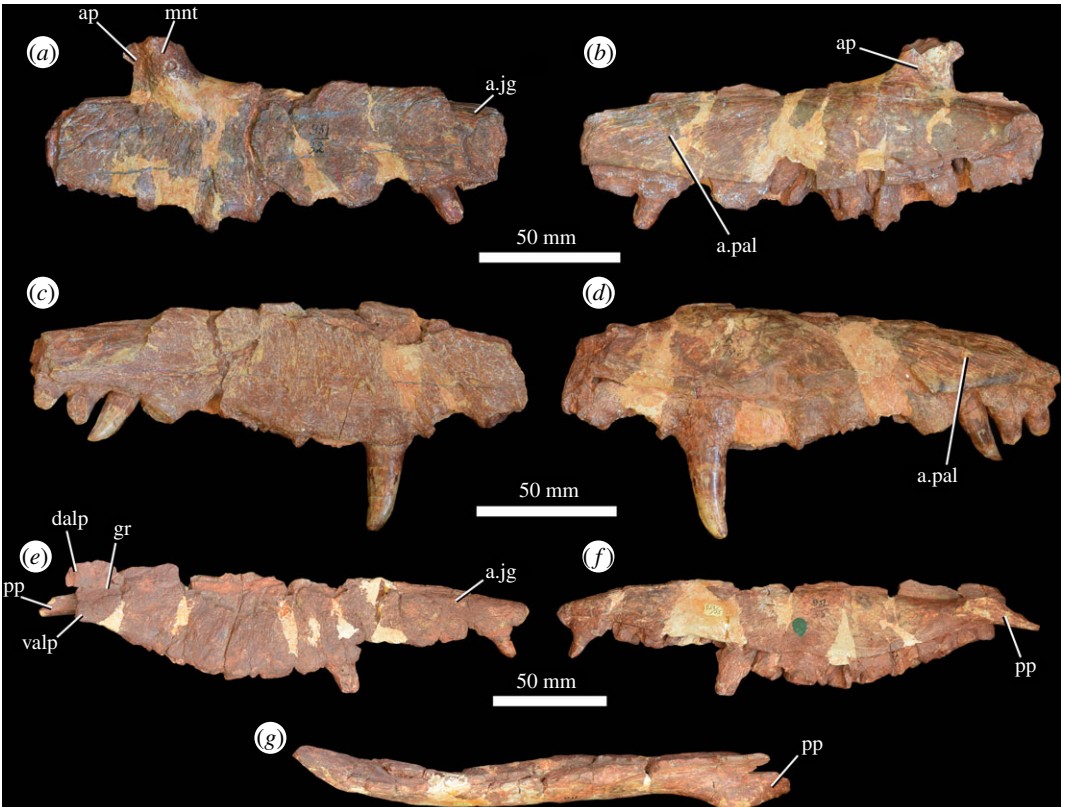

**Figure 4.** Maxilla of *G. prima* (='*V. triplicostata*'). Left maxilla (PIN 951/32) in lateral (*a*) and medial (*b*) views. Right maxilla (PIN 951/34) in lateral (*c*) and medial (*d*) views. Left maxilla (PIN 951/55) in lateral (*e*), medial (*f*) and dorsal (*g*) views. a.jg, articular surface for the jugal; ap, ascending process; a.pal, articular surface for the palatine; dalp, dorsal anterolateral process; gr, groove; mnt, maxillo-nasal tuberosity; pp, palatal process; valp, ventral anterolateral process.

the surrounding bone of the premaxilla (ankylothecodont implantation), contrasting with the non-ankylosed premaxillary teeth in the holotype of *G. prima* ([10]; PIN 2394/5). The replacement tooth does not appear to be fused to the surrounding socket. The two fully erupted teeth are incomplete, but are labiolingually compressed and recurved. The distal surfaces of the crowns are finely serrated, but the mesial margins are not sufficiently well preserved to assess the presence or absence of serrations.

## 5.2. Maxilla

Three partial maxillae are preserved (figure 4), two from the left side (PIN 951/32, PIN 951/55) and one from the right (PIN 951/34). All three maxillae lack their anterior and posterior ends and their ascending process, and in PIN 951/32 and PIN 951/34, the palatal process is also absent. All three bones are damaged along the margin of the antorbital fenestra. A few teeth are preserved *in situ*, but the dentition is generally either missing or badly damaged. The alveolar margin is damaged in all three elements, but appears to have been ventrally convex in lateral view and curved upwards towards the contact with the premaxilla (figure 4*c,e*), as is the case in other erythrosuchids [7]. The anterior end of the maxilla is best preserved in PIN 951/55, where it is divided into two anterior processes (figure 4*e*), although both are broken. A wide, shallow, anteroposteriorly extending groove lies on the lateral surface between these two processes (figure 4*e*: gr), as in the holotypes of *G. prima* (PIN 2394/5; [10]) and *Chalishevia cothurnata* (PIN 4366/1; [13]).

The lateral surface of the maxilla is gently sculpted by raised bosses, as occurs in the holotype of *G. prima* (PIN 2394/5; [10]). A row of foramina runs parallel to and approximately 1 cm above the alveolar margin. These foramina typically open ventrally and grooves extend ventrally from them, resembling the condition in other erythrosuchids [7]. The base of the ascending process is preserved in PIN 951/32 (figure 4*a*). On the lateral surface, there is a well-developed maxillo-nasal tuberosity (figure 4*a*: mnt). This tuberosity curves posteriorly towards the base of the ascending process, as is the case in other

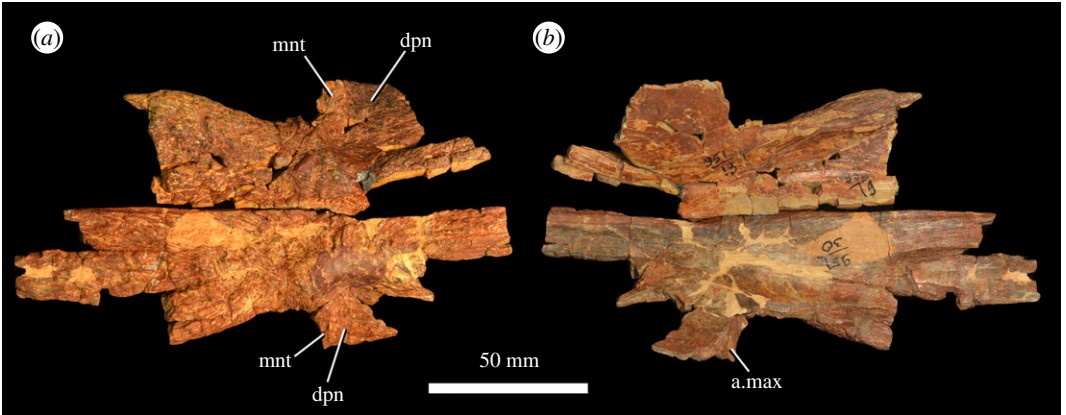

**Figure 5.** Isolated left (PIN 951/19-1) (*a*) and right (PIN 951/19-2) (*b*) nasals of *G. prima* (='*V. triplicostata*'), shown in inferred articulation and probably representing a single individual. a.max, articular surface for the maxilla; dpn, descending process; mnt, maxillo-nasal tuberosity.

erythrosuchids with the exception of *Fugusuchus hejiapanensis* and *Guchengosuchus shiguaiensis* [7,12]. A striated facet for articulation with the jugal is present on the lateral surface, immediately above the three most posteriorly preserved tooth sockets (figure 4*a*,*c*,*e*: a.jg). The anterior process of the jugal also articulated with a slot on the posterodorsal edge of the maxilla, as in other erythrosuchids (e.g. [9]).

Medially, there is a well-preserved, anteriorly projecting palatal process in PIN 951/55 (figure 4*e–g*). Its medial surface is grooved and rugose. The anterodorsal surface of the palatal process is bevelled where it was overlain dorsally by the palatal process of the premaxilla. On the medial surface of the maxilla, there is a strongly rugose area beginning above the sixth tooth and extending posteriorly until the end of the tooth row (figure 4*b*,*d*: a.pal). This probably represents the articular surface for the palatine. In ventral view, the maxilla curves laterally at its posterior end towards the contact with the jugal.

Eleven tooth positions are preserved in PIN 951/32, 12 positions in PIN 951/34 and 13 in PIN 951/55, which seems to have the complete tooth row preserved. This latter tooth count is consistent with that present in the holotype of *G. prima* (PIN 2394/5; [10]). At the very anterior end of the tooth row, there is a short edentulous region (diastema) along the ventral margin of the lower of the two anterior processes. In some cases, teeth are clearly fused to surrounding bone (ankylothecodont implantation), contrasting with the absence of fusion between the teeth and bone in the holotype of *G. prima* (PIN 2394/5; [10]). However, in some teeth of '*V. triplicostata*', this fusion is less clear and the tooth implantation appears thecodont (e.g. the ninth tooth of PIN 951/34). Very few teeth are well preserved, but the best-preserved teeth are labiolingually compressed, recurved and have fine serrations along both mesial and distal margins.

## 5.3. Nasal

Isolated left (PIN 951/19-1) and right (PIN 951/19-2) nasals are preserved, probably from a single individual (figure 5). Parts of both nasals are also preserved in the '*V. triplicostata*' lectotype skull roof (PIN 951/59; figure 6). All of the nasals are incomplete. The nature of the anterior contact with the premaxilla is unknown because all are broken anteriorly. Laterally, the nasal forms an anteroposteriorly straight contact with the prefrontal. The nasal is elongate and extends posteriorly to contact the frontal above the anterodorsal margin of the orbit. Anteriorly, the dorsal surface of the paired nasals is strongly transversely convex. Level with the antorbital fenestrae, the paired nasals bear a transversely broad median fossa that covers almost the entire dorsal surface of the bones (figure 6*a*). This fossa has been interpreted as an autapomorphy of *G. prima* [10]. Posterior to the fossa, the dorsal surface of the paired nasals becomes mostly flat, although there is a transversely narrow, anteroposteriorly elongate fossa on the midline that extends onto the paired frontals. The dorsal surface of the nasal is rugose. The lateral surface of the nasal is generally poorly preserved, but anteriorly there is a descending process placed towards the anterior end of the bone (figures 5 and 6: dpn), the posterior margin of which bears a facet for articulation with the ascending process of the maxilla (figure 5*b*: a.max). The posterior edge of the descending process is raised into a ridge that formed the dorsal portion of the maxillo-nasal tuberosity (figure 5*a*: mnt). The lateral margin of the nasal is dorsoventrally thickened posteriorly and would have formed the dorsal border of the

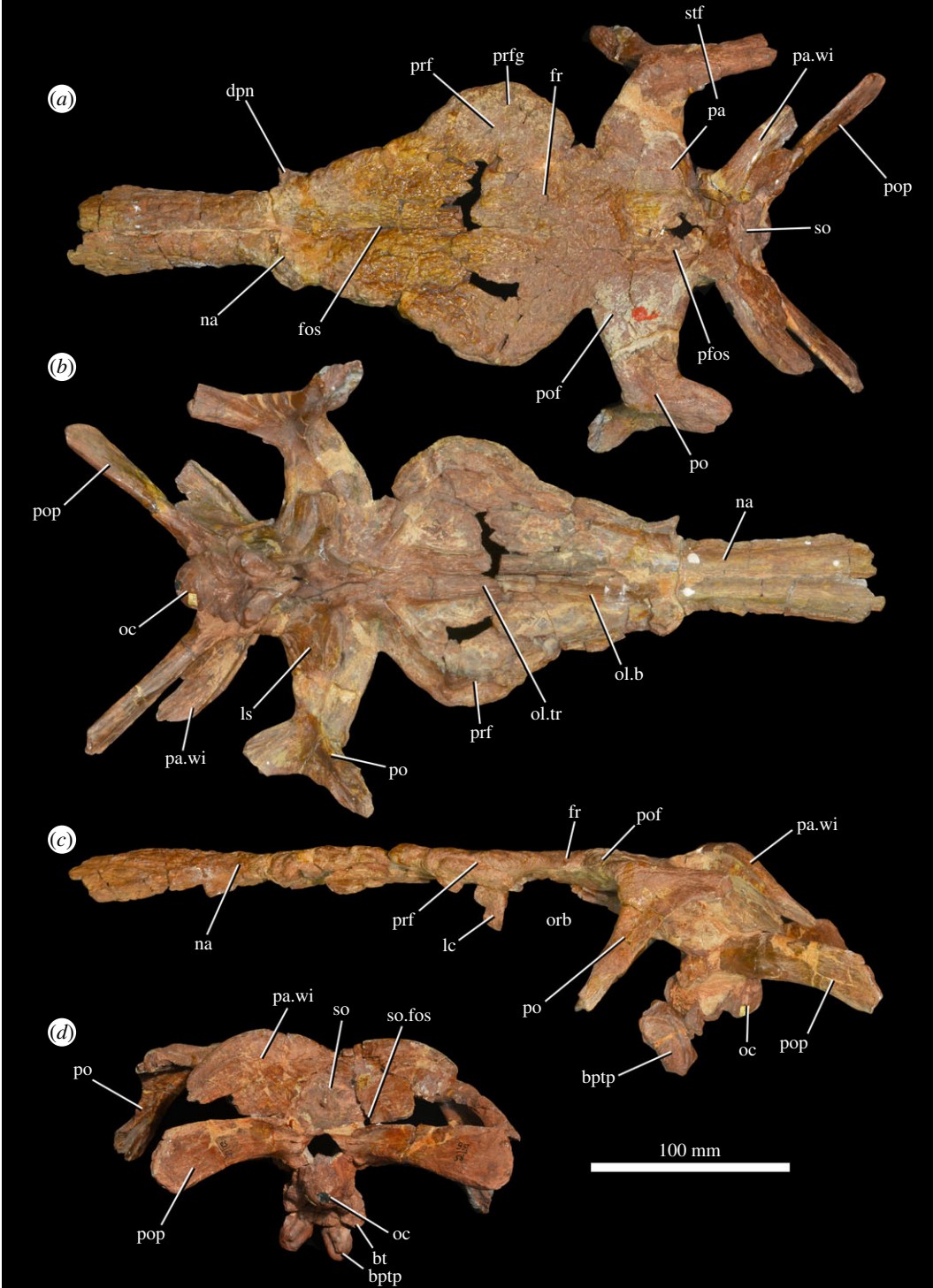

**Figure 6.** Skull roof and braincase of *G. prima* (PIN 951/59; lectotype of '*V. triplicostata*') in dorsal (*a*), ventral (*b*), left lateral (*c*) and posterior (*d*) views. bptp, basipterygoid process; bt, basal tuber; dpn, descending process of nasal; fos, fossa on dorsal surface of nasals and frontals; fr, frontal; lc, lacrimal; ls, laterosphenoid; na, nasal; oc, occipital condyle; ol.b, olfactory bulb; ol.tr, olfactory tract; orb, orbit; pa, parietal; pa.wi, parietal wing; pfos, pineal fossa; po, postorbital; pof, postfrontal; pop, paroccipital process; prf, prefrontal; prfg, groove on dorsal surface of prefrontal; so, supraoccipital; so.fos, fossa on supraoccipital; stf, supratemporal fenestra.

antorbital fossa. On the ventral surface of the posterior end of the paired nasals, the impressions of the olfactory tract and bulbs are preserved (figure 6*b*). The olfactory tract would have been transversely narrow. The margins of the olfactory bulbs are poorly delimited but also appear to be transversely narrow. The medial surfaces of the nasals articulate with each other with ridge-in-groove articulations.

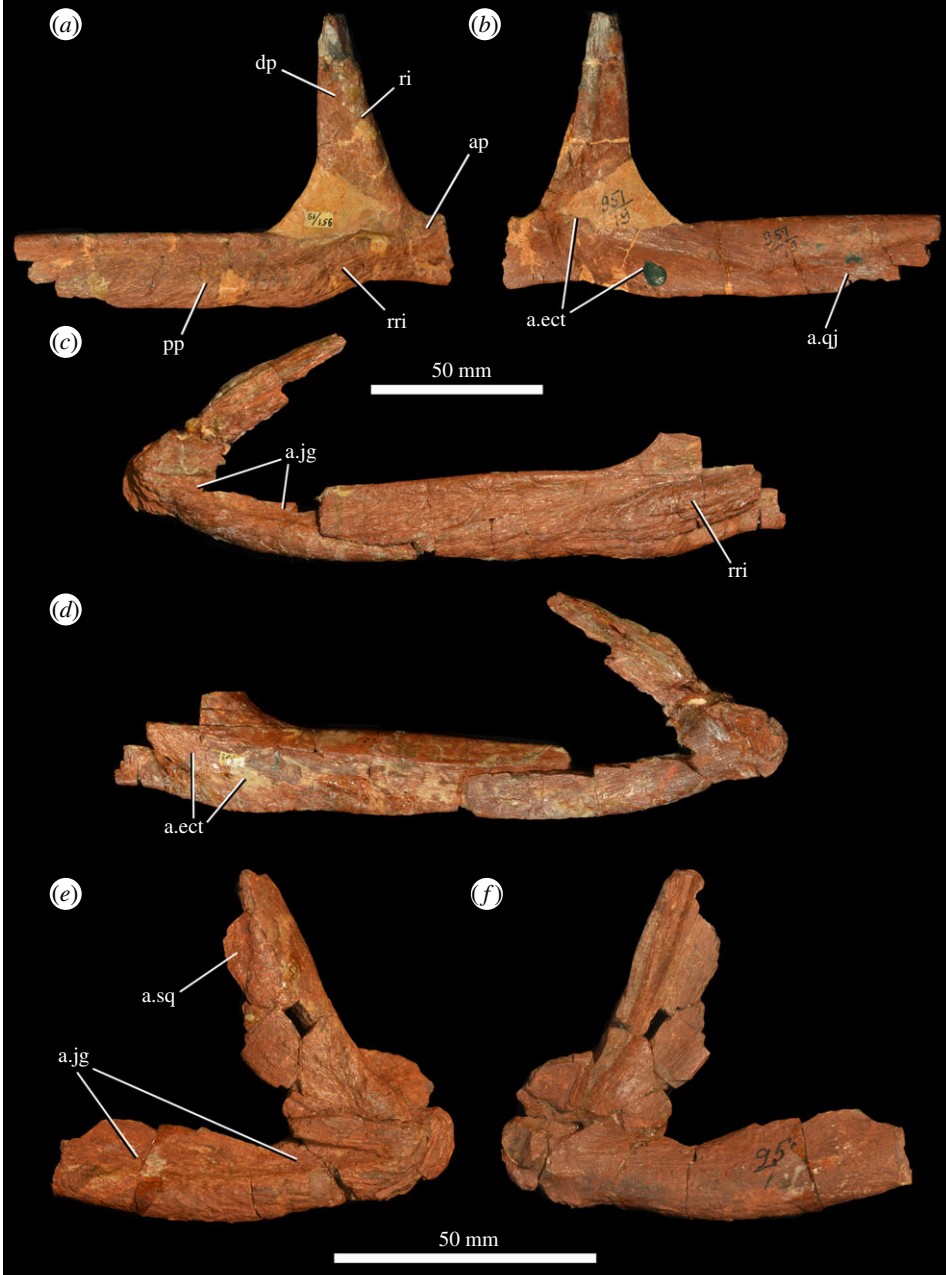

**Figure 7.** Jugal and quadratojugal of G. prima (='V. triplicostata'). Right jugal (PIN 951/117) in lateral (a) and medial (b) views. Articulated right jugal and quadratojugal (PIN 951/23-2) in lateral (c) and medial (d) views. Left quadratojugal (PIN 951/17-2) in lateral (e) and medial (f) views. a.ect, articular surface for the ectopterygoid; a.jg, articular surface for the jugal; ap, anterior process of jugal; a.qj, articular surface for the quadratojugal; a.sq, articular surface for the squamosal; dp, dorsal process of jugal; pp, posterior process of jugal; ri, ridge on lateral surface of dorsal process; rri, strongly rugose ridge.

## 5.4. Lacrimal

Only fragments of the lacrimal appear to be preserved, and they are in articulation with the left prefrontal of the 'V. triplicostata' lectotype (PIN 951/59; figure 6c). These fragments do not provide any notable anatomical information. Von Huene (pl. 11, fig. 3 in [16]) figured what he identified as a partial right lacrimal and prefrontal, but we have not been able to locate this among the available material.

## 5.5. Jugal

Parts of four jugals are preserved (figure 7a–d). PIN 951/117 is a right jugal missing the anterior process and the tips of the dorsal and posterior processes and is reconstructed in places (figure 7a,b). PIN 951/23-1 is a left jugal missing the anterior process, badly damaged at the junction between the dorsal and

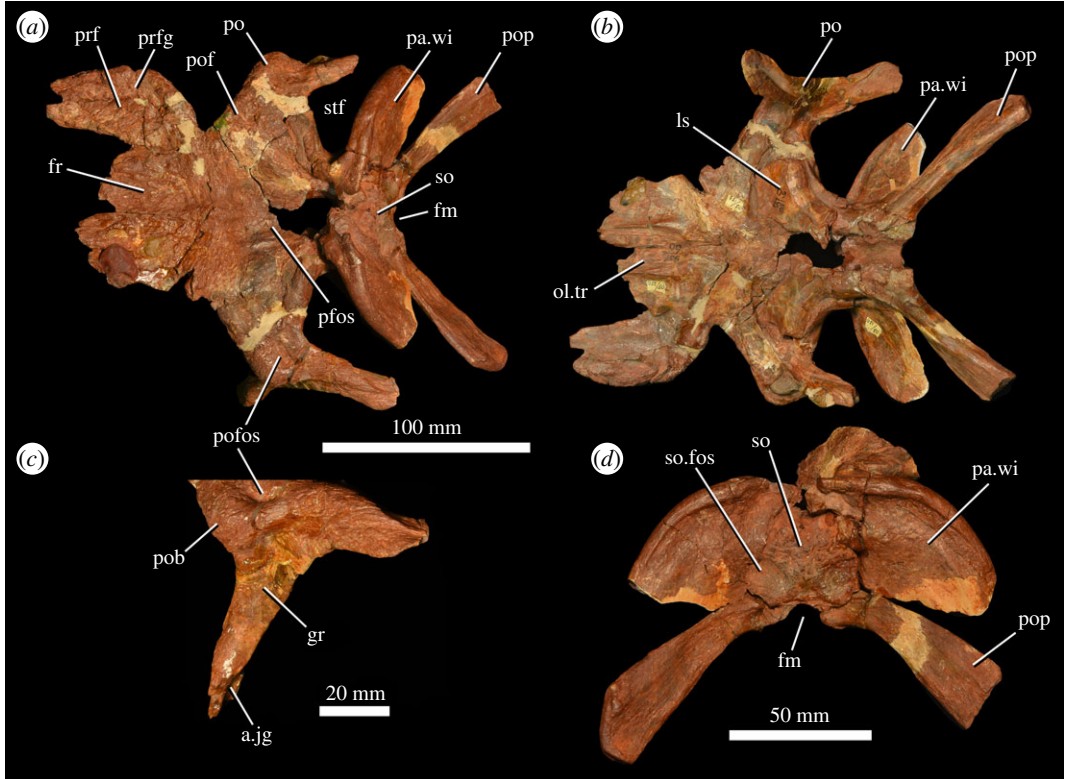

**Figure 8.** Partial skull roof and braincase (PIN 951/60) of *G. prima* (='*V. triplicostata*') in dorsal (*a*) and ventral (*b*) views, close-up of left postorbital in lateral view (*c*) and posterior view (*d*). a.jg, articular surface for the jugal; fm, foramen magnum; fr, frontal; gr, groove on lateral surface of ventral process of postorbital; ls, laterosphenoid; ol.tr, olfactory tract; pa.wi, parietal wing; pfos, pineal fossa; po, postorbital; pob, postorbital boss; pof, postfrontal; pofos, fossa on dorsal surface of postorbital; pop, paroccipital process; prf, prefrontal; prfg, groove on dorsal surface of prefrontal; so, supraoccipital; so.fos, fossa on supraoccipital; stf, supratemporal fenestra.

posterior processes, both of which are missing their tips. Another right jugal, PIN 951/23-2 (figure 7*c,d*), is preserved in articulation with the right quadratojugal and is missing its anterior and dorsal process and the posterior end of the posterior process. PIN 951/9 is a left jugal missing its dorsal process and most of the anterior process.

The jugal is triradiate and formed the ventral margin and lower part of the posterior margin of the orbit. It also probably formed about half of the anterior margin of the infratemporal fenestra, and the entire ventral margin of the infratemporal fenestra, extensively overlapping the anterior process of the quadratojugal. The dorsal process has a rugose ridge on its lateral surface adjacent to the orbital margin (figure 7*a*). The anterior surface of the process is grooved for articulation with the ventral process of the postorbital. Medially, the anterior part of the base of the dorsal process is recessed laterally to form an articular facet for the ectopterygoid (figure 7*b,d*: a.ect); this facet extends posteroventrally onto the base of the posterior jugal process. The main body of the jugal beneath the dorsal process is strongly thickened transversely on its lateral surface forming a rugose, anterodorsally to posteroventrally oriented ridge (figure 7*a,c*: rri), as also occurs in the holotype of *G. prima* (PIN 2394/5; [10]). The posterior process of the jugal is elongate. The distal end of the posterior process tapers and fits into a notch on the lateral surface of the quadratojugal (figure 7*c,e*). Medially, there is an anteroposteriorly elongate, striated and dorsoventrally concave facet on the posterior process for the articulation with the quadratojugal (figure 7*b*: a.qj).

## 5.6. Prefrontal

Both prefrontals are preserved in the '*V. triplicostata*' lectotype (PIN 951/59; figure 6), and the right prefrontal is preserved in PIN 951/60 (fig. 3 in [18]), an articulated skull roof (figure 8). In both specimens, the majority of the dorsal surface of each prefrontal is preserved, but the ventral surfaces are damaged. The prefrontal has a straight medial margin in dorsal and ventral views and forms an

extensive contact with both the frontal and the nasal. The lateral margin of the prefrontal is strongly convex in dorsal view (figures 6*a* and 8*a*). The prefrontal is slightly thickened dorsoventrally along its lateral margin relative to the more medial part of the skull table and would have overhung the lacrimal. Immediately medial to this lateral thickening, there is an anteroposteriorly extending, shallow curved groove on the skull table (figures 6*a* and 8*a*: prfg), resembling the condition in the holotype of *G. prima* (PIN 2394/5; [10]). The dorsal surface of the prefrontal is strongly rugose. The anterolateral margin of the bone has a well-developed groove (figure 6*c*) that extends diagonally across the bone from the anterodorsal corner to the posteroventral corner, as occurs in the holotypes of *G. prima* (PIN 2394/5; [10]) and *E. africanus* [9], and a referred specimen of *G. madiba* [11].

## 5.7. Frontal

The frontals are preserved in the '*V. triplicostata*' lectotype (PIN 951/59; figure 6) and in PIN 951/60, the articulated skull roof (figure 8). The paired frontals contact the nasals anteriorly (although this is broken in PIN 951/60) and the parietals posteriorly, and each frontal forms contacts with the prefrontal and postfrontal laterally. The frontal does not contact the postorbital. The frontal forms a small part of the dorsal rim of the orbit, as in the holotype of *G. prima* (PIN 2394/5; [10]). In other erythrosuchids, the frontals have a similarly small participation or do not contribute to the border of the orbit, a condition that it is intraspecifically variable in some species [9,10,38]. Anteriorly, the dorsal surface of the paired frontals is flattened, but bears a narrow median fossa that is continuous with the median fossa on the nasals (figure 6*a*: fos). This fossa has been proposed as an autapomorphy of *G. prima* [10]. It appears to be less well developed in '*V. triplicostata*' than in the type specimen of *G. prima* (PIN 2394/5), but the depth of the fossa may have been exaggerated in the latter due to transverse compression of the skull. This fossa terminates posteriorly slightly anterior to the point at which the skull roof is transversely narrowest in dorsal view. Posteriorly, the contacts with the parietal and postfrontal are not very clear due to poor preservation, but some interdigitation is visible. The posterior contact with the parietal appears to be transversely straight and extends through the pineal fossa. The paired frontals form the anterior 40% of the pineal fossa (figures 6*a* and 8*a*: pfos), which is dorsoventrally deep with a subrectangular outline in dorsal view, resembling the conditions in the holotype of *G. prima* and *E. africanus* [9,10]. The dorsal surface of the frontal is generally ornamented with low rugosities, but is smooth within the pineal fossa. Ventrally, the frontal is broadly contacted by the laterosphenoid and impressions of the roof of the endocranial cavity and the olfactory tract are present (figures 6*b* and 8*b*).

## 5.8. Postfrontal

The postfrontals are present in both the '*V. triplicostata*' lectotype (PIN 951/59; figure 6) and the other articulated skull roof (PIN 951/60; figure 8), but their contacts with the frontals, postorbitals and parietals are difficult to identify with certainty. The postfrontal is a relatively small element that forms a small posterodorsal part of the orbital rim. Ventrally, it probably formed at least a small contact with the laterosphenoid. A shallow pit is present on the dorsal surface of the postfrontal, close to its contact with the frontal.

## 5.9. Postorbital

Right and left postorbitals are present in the '*V. triplicostata*' lectotype (PIN 951/59; figure 6) and the other articulated skull roof (PIN 951/60; figure 8), but are generally broken at the ends of their ventral and posterior processes. The postorbital is a triradiate bone. It has an elongate ventral process that formed much of the posterior margin of the orbit (figure 8*c*). There is a deep groove on the lateral surface on the dorsal part of this process (figure 8*c*: gr). Ventrally, the process twists and is transversely expanded to form a flattened and heavily striated, anteriorly facing surface. Medially, there is a deep, elongate groove to receive the dorsal process of the jugal (figure 8*c*: a.jg). The groove on the lateral surface of the ventral process arches posteriorly and extends onto the lateral surface of the posterior process. It is continuous with the groove on the lateral surface of the squamosal. The posterior postorbital process is similarly sized in transverse width and dorsoventral height. It has a medial flange that fits into the groove on the anterior process of the squamosal. The dorsal surface of the postorbital is rugose and ornamented. It contacts the parietal and postfrontal medially and forms the anterior part of the lateral margin of the supratemporal fenestra as well as a small part of the

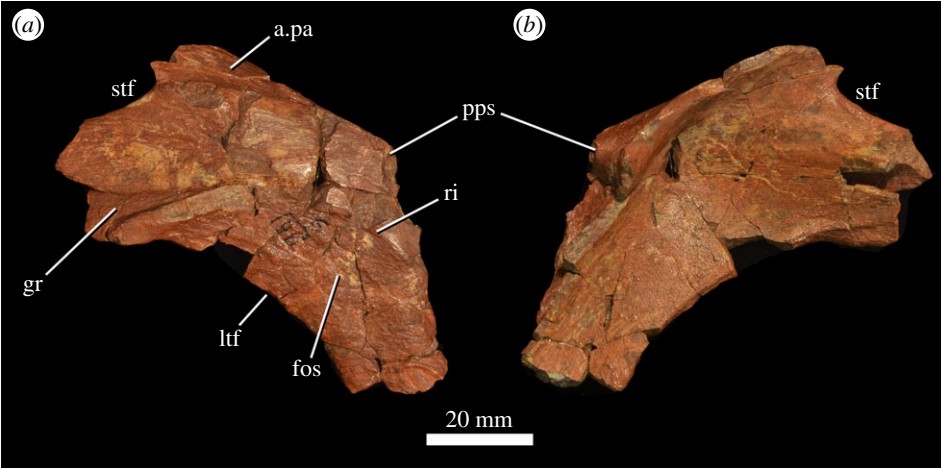

**Figure 9.** Left squamosal (PIN 951/118) of *G. prima* (='*V. triplicostata*') in lateral (*a*) and medial (*b*) views. a.pa, articular surface for the parietal; fos, fossa adjacent to the border of the infratemporal fenestra; ltf, border of the infratemporal fenestra; pps, broken base of the posterior process of the squamosal; ri, ridge delimiting fossa on the ventral process of the squamosal; stf, border of the supratemporal fenestra.

anterior part of this fenestra. There is a well-developed fossa on the dorsal surface of the postorbital above the ventral process (figure 8*c*: pofos), as occurs in the holotype of *G. prima*, *G. madiba* and *E. africanus* [10]. A strongly rugose boss is present on the orbital margin at the point where the ventral and medial processes meet (figure 8*c*: pob). This boss slightly overhangs the ventral process and has a shallow groove present on its lateral surface. This groove is present but slightly less well developed in the type specimen of *G. prima* (PIN 2394/5; [10]).

## 5.10. Squamosal

There are three partial squamosals preserved, two from the left side (PIN 951/118, 119; figure 9) and one from the right (PIN 951/120). They are all broken along their anterior, medial, ventral and posterior margins. The base of the ventral process is recessed medially relative to the rest of the process and would have formed a fossa adjacent to the posterodorsal corner of the infratemporal fenestra (figure 9*a*: fos). This fossa is delimited by a low ridge that extends onto the anterior process. Dorsal to this depression is a deep elongate groove on the lateral surface of the anterior process (figure 9*a*: gr), which received the posterior process of the postorbital. Medial to the groove is a flattened surface that borders the posterolateral corner of the supratemporal fenestra. More medially there is a heavily striated surface that would have articulated with the posterolateral wing of the parietal (figure 9*a*: a.pa). Posterior to this striated surface, there is a medially and slightly posteriorly facing smooth surface that articulated with the anterolateral surface of the distal end of the paroccipital process.

## 5.11. Quadratojugal

Three quadratojugals are present (figure 7*c–f*). PIN 951/23-2 is a right quadratojugal preserved in articulation with a right jugal (figure 7*c,d*), and PIN 951/17-2 (figure 7*e,f*) and PIN 951/17-1 are isolated left quadratojugals. None of the quadratojugals are complete, and all lack some or most of the anterior process and the dorsal part of the dorsal process. The anterior process is elongate, extending slightly beyond the mid-length of the infratemporal fenestra. It is strongly medially recessed dorsolaterally where it was overlapped extensively by the jugal (figure 7*e*: a.jg), which excluded the quadratojugal entirely from the ventral border of the infratemporal fenestra. This articular surface for the jugal is striated. The dorsal process of the quadratojugal is anterodorsally directed, meaning that the posteroventral corner of the infratemporal fenestra would have formed an acute angle, as occurs in the holotype of *G. prima* (PIN 2394/5; [10]). The posteroventral corner of the bone is very strongly rugose adjacent to the contact with the quadrate condyles, resembling the condition in *E. africanus* [9], *G. madiba* [11] and the holotype of *G. prima* (PIN 2394/5; [10]). In PIN 951/17-2, part of the articular facet for the ventral process of the squamosal is preserved (figure 7*e*: a.sq), suggesting that this process of the squamosal would have extended along most, but not all, of the posterior

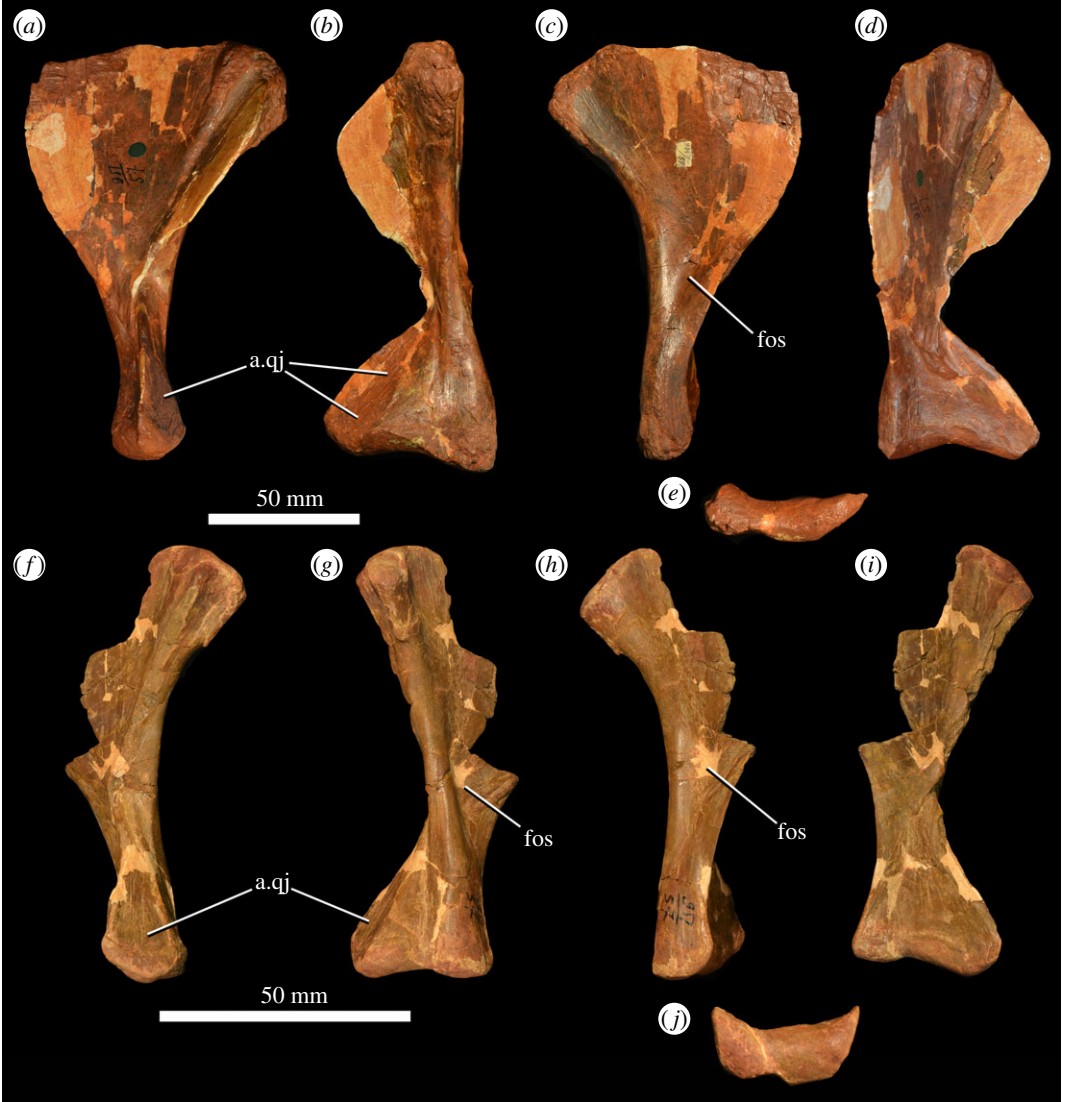

**Figure 10.** Two left quadrates (PIN 951/57-1: (a–e); PIN 951/57-3: (f–j)) of *G. prima* (=*'V. triplicostata'*) in lateral (a,f), posterior (b,g), medial (c,h) and anterior (d,i) views, and outline of quadrate condyles in ventral view (e,j). a.qj, articular surface for quadratojugal; fos, fossa on pterygoid wing of quadrate.

margin of the infratemporal fenestra. This is similar to the condition in the holotype of *G. prima* [10], but contrasts with the condition in *E. africanus* and *S. shansisuchus* in which all or nearly all of the posterior margin of the infratemporal fenestra is formed by the squamosal [7,9,38,55].

## 5.12. Quadrate

There are three quadrates preserved, all from the left side (PIN 951/57-1, PIN 951/57-2, PIN 951/57-3; figure 10). The condyles and heads are well preserved in all three quadrates, but the lateral flanges and the pterygoid wings are all damaged. In lateral view, the posterior margin is strongly concave along its length (figure 10a,f). There is a broad, flattened and gently striated surface for articulation with the quadratojugal on the lateral margin of the ventral part of the quadrate (figure 10a,b,f,g: a.qj). The medial margin of the quadrate foramen is not well preserved. There is a groove on the posterior surface of the bone that extends ventrally down the middle of the shaft from the waisted midshaft region towards the condyles. The quadrate head is transversely thickened relative to the shaft of the bone, and convex in lateral view. Its articular surface is rugose. The quadrate condyles are transversely broad and anteroposteriorly narrow, forming a saddle-shaped articular surface (figure 10e,j). In medial view, there is a fossa on the base of the pterygoid wing (figure 10c,h: fos).

The quadrate condyles of the holotype of *G. prima* (PIN 2394/5) are not as transversely expanded and are more asymmetrically developed anteriorly, but this probably results from transverse crushing of PIN 2394/5.

## 5.13. Parietal

The parietals are present in both the '*V. triplicostata*' lectotype (PIN 951/59; figure 6) and the skull roof specimen (PIN 951/60; figure 8). In both cases, the central part of the parietal region is not well preserved, so the suture between the parietals cannot be identified. The paired parietals form the posterior 60% of the pineal fossa (figures 6 and 8: pfos) and have anterolateral processes that contact the postorbital and postfrontal. The parietal forms the majority of the posterior margin of the supratemporal fenestra, the entirety of the medial margin of this fenestra and part of its anterior margin. It cannot be determined if a pineal foramen was present. The parietal has a very well-developed posterolateral wing (figures 6 and 8: pa.wi), which is plate-like and strongly dorsally convex in posterior view, as occurs in other erythrosuchids [7]. The dorsal part of this wing is anteroposteriorly thickened, slightly overhanging the occiput. This thickened dorsal margin bears a median groove along its dorsal surface. The posterolateral wings are strongly downturned at their distal ends and ventrally contact the paroccipital processes. Medially, the parietals contact the supraoccipital on the occiput. On the anterior surface of the parietal wing, there is a well-developed, curved ridge. The distal part of the wing is striated for articulation with the squamosal. There is no evidence that a separate interparietal was present.

## 5.14. Braincase

The braincase is preserved in both the '*V. triplicostata*' lectotype (PIN 951/59; figure 6) and the other skull roof specimen (PIN 951/60; figure 8). It has been described in detail by Gower & Sennikov [25], and we do not provide further descriptive notes here, although we do describe the supraoccipital, which was not described by Gower & Sennikov [25]. The supraoccipital is a subtriangular element that articulates laterally and dorsally with the parietal and ventrolaterally with the otoccipital (figures 6a,d and 8a,d: so). It forms at least a small component of the dorsal margin of the foramen magnum (contra [25,26]). There are large paired fossae on the ventral part of the supraoccipital that each extend onto the adjacent lateral part of the posterolateral wing of the parietal (figures 6d and 8d: so.fos). This region of the occiput is heavily deformed, damaged and partly reconstructed in the holotype of *G. prima* (PIN 2394/5) so the presence or absence of these fossae cannot be confirmed in that specimen [10].

## 5.15. Pterygoid

There is a relatively complete left pterygoid (PIN 951/15-1; figure 11) and a fragment of a right pterygoid (PIN 951/15-2). The pterygoid is missing its quadrate wing and the articular surface for the basipterygoid. The dorsal process is broken along its margins. The posteroventral process is well developed (figure 11a,b: pvp), with a broad, prominent ridge extending along its ventral surface (figure 11a: pvp.r). The anterior process is broken anteriorly, but has two ridges, a midline ridge (figure 11a: ap.mr) and an anterolaterally directed ridge (figure 11a: ap.alr), as occurs in the holotype of *G. prima* (PIN 2394/5; [10]). Between the latter ridge and the ridge on the posteroventral process, the ventral surface of the pterygoid is strongly concave. Several very small palatal teeth are present on the two ridges of the anterior process close to the point at which they converge with one another (figure 11c). Similar teeth are not clearly present in the holotype of *G. prima* (PIN 2394/5), although this area is not well preserved in that specimen. Palatal teeth are absent in other erythrosuchids [7,9]. There are some possible scattered, very small palatal teeth on the base of the ridge on the posteroventral process in PIN 951/15-2, but these do not appear to be present in PIN 951/15-1. These two pterygoids may represent a single individual based on size, morphology and preservation, potentially suggesting that there is some variation in the presence or absence of these teeth, even within the same individual. Such variation has not previously been documented in other erythrosuchids.

## 5.16. Palatine

A partial right palatine (PIN 951/18-1; figure 12) and a somewhat less complete left palatine (PIN 951/18-2) are preserved. The former element was described by Ochev [22], although he interpreted it as being from the left side, rather than the right as interpreted here. The ventral surface of the palatine has a broad,

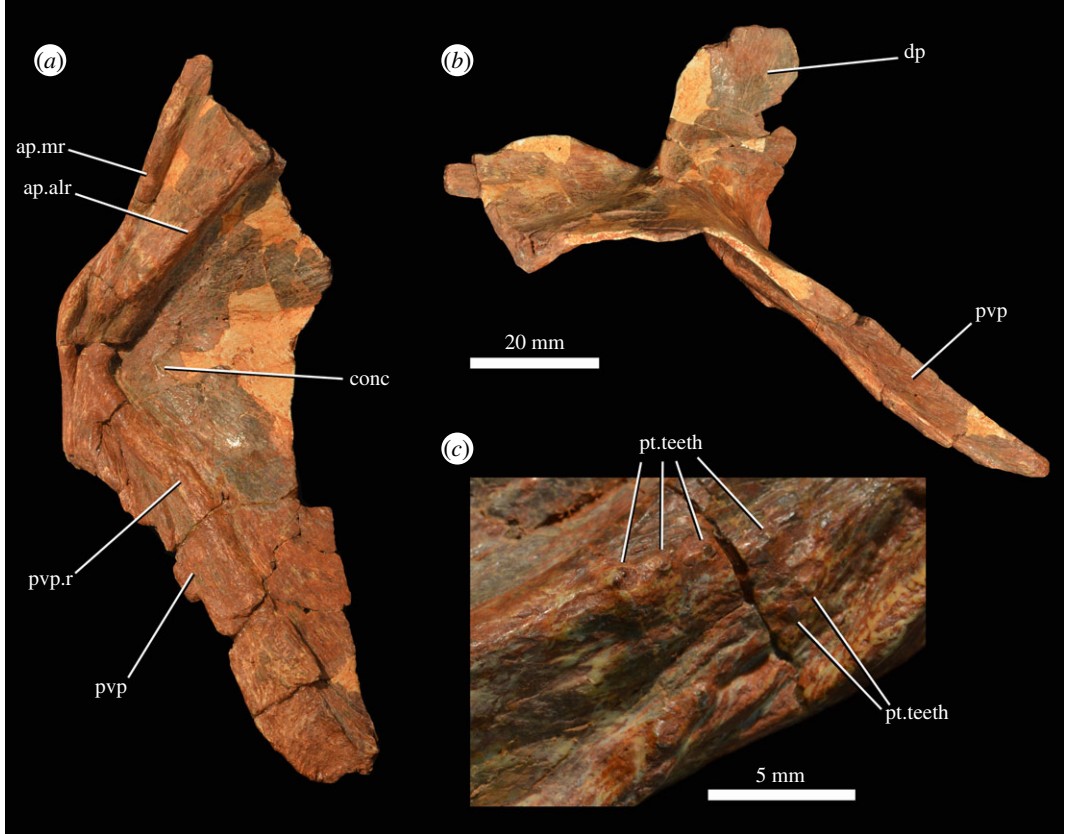

**Figure 11.** Left pterygoid (PIN 951/15-1) of *G. prima* (='*V. triplicostata*') in ventral (*a*) and lateral (*b*) views, with close-up of the bases of the medial and anterolateral ridges on the anterior process (*c*), showing pterygoid teeth. ap.alr, anterolateral ridge on anterior process; ap.mr, medial ridge on anterior process; conc, concavity; dp, dorsal process; pt.teeth, pterygoid teeth; pvp, posteroventral process; pvp.r, ridge on posteroventral process.

transversely thick ridge (figure 12*a*: pal.r), which separates two concave regions. This ridge would have been continuous with the anterolaterally directed ridge on the anterior process of the pterygoid, as occurs in the holotype of *G. prima* (PIN 2394/5; [10]). There are some possible, scattered, very small teeth on this ridge on the right palatine (figure 12*d*). Preservation of the ridge on the left element is too poor to determine the presence or absence of these teeth. There is a straight lateral margin that would have contacted the maxilla, with a broad, striated triangular surface for contact (figure 12*c*: a.mx). As noted by Ochev [22], a shallow groove ('sulcus 1' of [22]) crosses this surface close to its posterior end (figure 12*c*: gr). Anterior and medial to the ridge on the ventral surface, the palatine is drawn out into a dorsomedially directed flange (figure 12*a*,*c*: dmf). There is a dorsally recessed area on the ventral surface of the flange, which would have been overlapped by the pterygoid. The dorsal surface of the palatine has a deep fossa adjacent to the contact with the maxilla (figure 12*b*: fos).

## 5.17. Dentary

A minimum of six dentaries are preserved (figures 13 and 14). These include a large right dentary with an articulated splenial (PIN 951/69-1) and fragment of left dentary (PIN 951/69-3), which were collected from the Rassypnaya locality in 1974, later than the other specimens; an anterior part of a right dentary (PIN 951/54-1); an anterior part of a left dentary (PIN 951/54-3, possibly from the same individual as the previous specimen); an anterior part of a left dentary with a partial splenial in articulation (PIN 951/30-1); and a largely complete right dentary with an articulated splenial (PIN 951/46-1, this might be the same individual as PIN 951/30-1 based on size and similar preservation). The posterior part of this latter dentary is preserved separately (PIN 951/46-2) in articulation with post-dentary bones (figure 15*a*,*b*). These two parts (PIN 951/46-1 and 951/46-2) together form almost a complete hemimandible. This hemimandible is almost identical in length to the complete right hemimandible of the holotype of *G. prima* (PIN 2394/5).

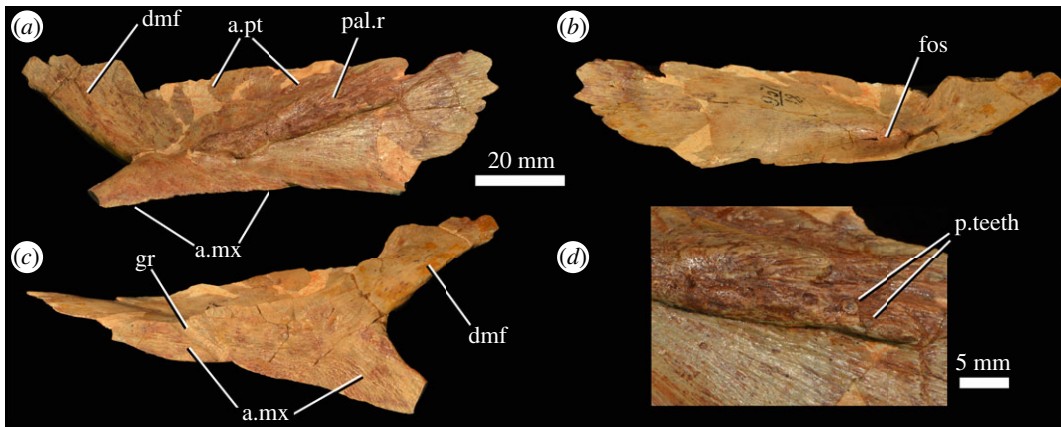

**Figure 12.** Right palatine (PIN 951/18-1) of *G. prima* (='*V. triplicostata*') in ventral (*a*), dorsal (*b*) and lateral (*c*) views, and with close-up of the base of the ridge on the ventral surface of the element (*d*), showing possible palatine teeth. a.mx, articular surface for the maxilla; a.pt, articular surface for the pterygoid; dmf, dorsomedial flange; fos, fossa; gr, groove extending across the articular surface for the maxilla; pal.r, ridge on ventral surface of the palatine; p.teeth, possible palatine teeth.

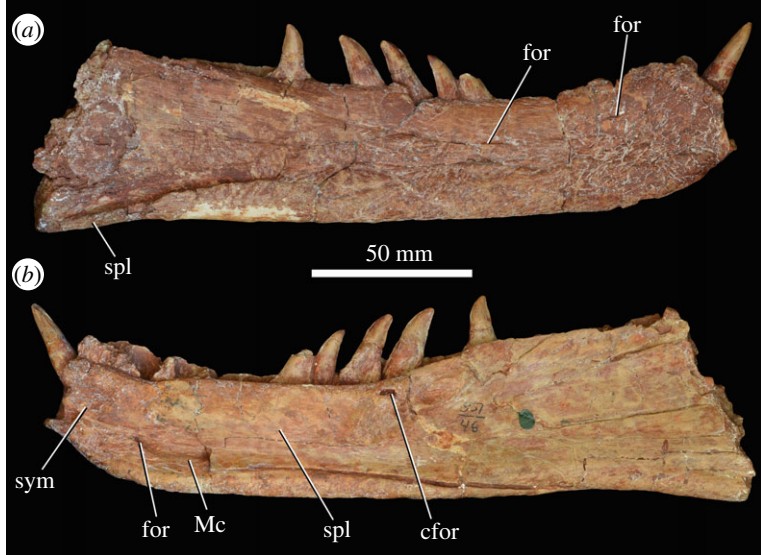

**Figure 13.** Right dentary (PIN 951/46-1) of *G. prima* (='*V. triplicostata*') in lateral (*a*) and medial (*b*) views. cfor, crater-like foramen, probably pathological; for, foramen; Mc, Meckelian canal; spl, splenial; sym, symphysis.

There appear to be 12–13 dentary teeth (13 in PIN 951/30-1 in which the entire tooth row appears to be preserved; figure 14*a–c*), resembling the condition in the holotype of *G. prima* (13–14 dentary teeth; PIN 2394/5; [10]) and in *Sarmatosuchus otschevi* (14 dentary teeth; [56]), *G. madiba* (14 dentary teeth; NM QR 3051) and *S. shansisuchus* (14–16; [38]). In lateral view, the ventral margin of the dentary is straight, and the dorsal margin is concave (figures 13*a* and 14*a*). It is expanded dorsally at both the posterior and anterior ends, which is a common condition among erythrosuchids [7,9,38]. The external surface of the bone is rugose. There are numerous foramina on the lateral surface, mostly occurring in a row that runs parallel to the element's dorsal edge, about 1 cm below the tooth margin (figures 13*a* and 14*a,d*: for). The lateral surface of the dentary is very gently concave dorsoventrally, being slightly thickened at its ventral margin. Medially, there is a laterally recessed symphysial surface positioned below to the first two tooth positions (figures 13*b* and 14*b,e*: sym). Posterior to the symphysis, there is a broad Meckelian canal that becomes deeper and dorsoventrally broader posteriorly. This canal is positioned on the ventral half of the bone. There is a foramen that opens posteriorly, which is positioned within the ventral part of this canal, beneath the third tooth position and just posterior to the symphyseal region (figures 13*b* and 14*b,e*).

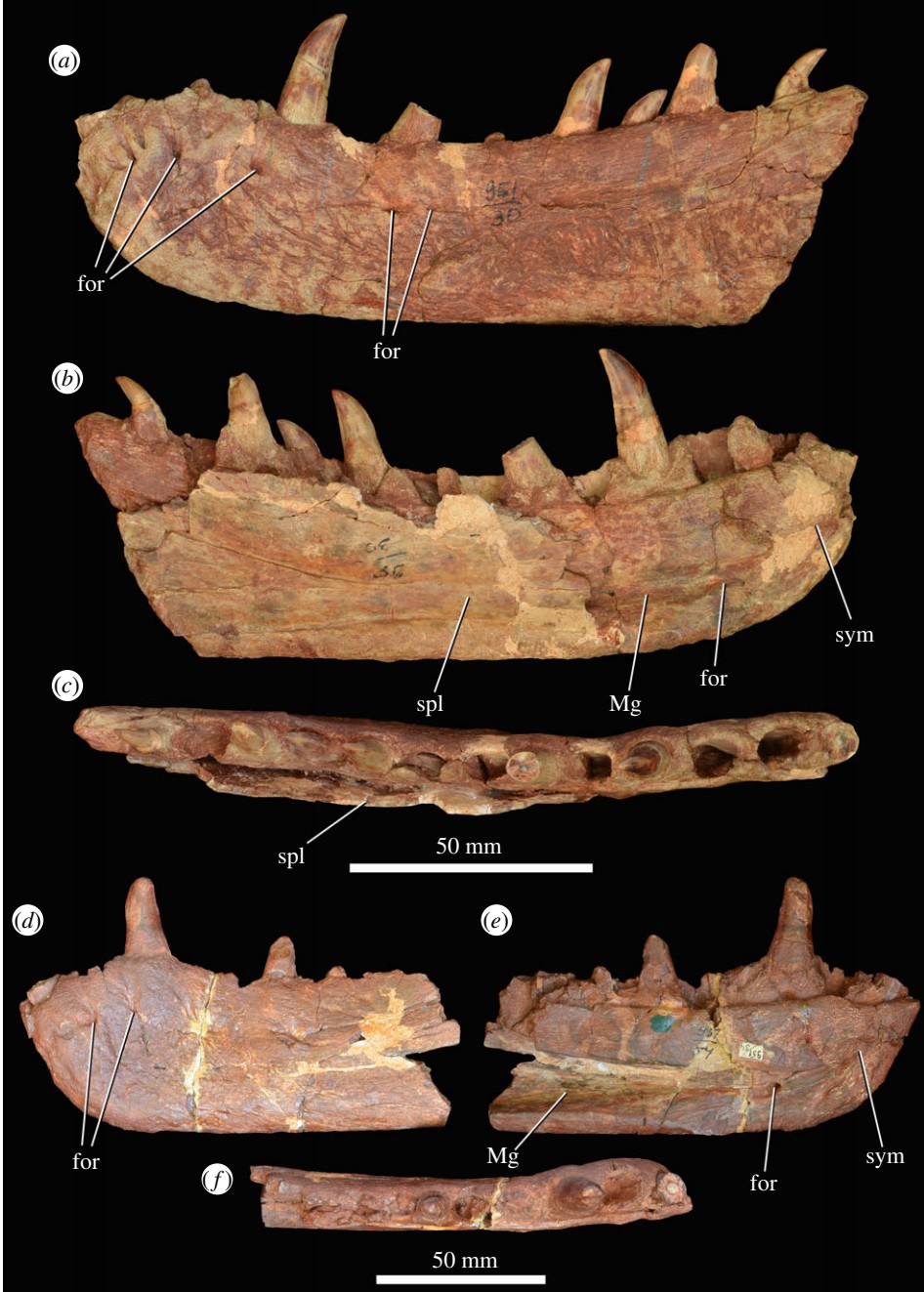

**Figure 14.** Left dentaries (PIN 951/30-1: (a–c); PIN 951/54-1: (d–f)) of *G. prima* (='*V. triplicostata*') in lateral (a,d), medial (b,e), and dorsal (c,f) views. for, foramen; Mc, Meckelian canal; spl, splenial; sym, symphysis.

The first dentary tooth is procumbent, whereas more posterior teeth are directed dorsally. The morphology of the first tooth position is unknown in the holotype of *G. prima* because of damage (PIN 2394/5; [10]). The teeth are labiolingually compressed and recurved, with serrations along both their mesial and distal margins, extending close to the base of the tooth on the distal margin. Some of the dentary teeth appear to be fused to surrounding bone (ankylothecodont implantation). In other cases, this fusion does not appear to have occurred and the implantation appears to be thecodont, as in the holotype of *G. prima* (PIN 2394/5; [10]). This variation can occur within a single tooth row (e.g. PIN 951/30-1).

The posterior end of the dentary is only preserved in PIN 951/46-2, in which a small part of the posterior end is preserved immediately anterior to the external mandibular fenestra (figure 15a,b). There appears to be a ventral process of the dentary, which articulates with the angular, and a central process (figure 15a: pcp), which extends along the dorsal margin of the external mandibular fenestra

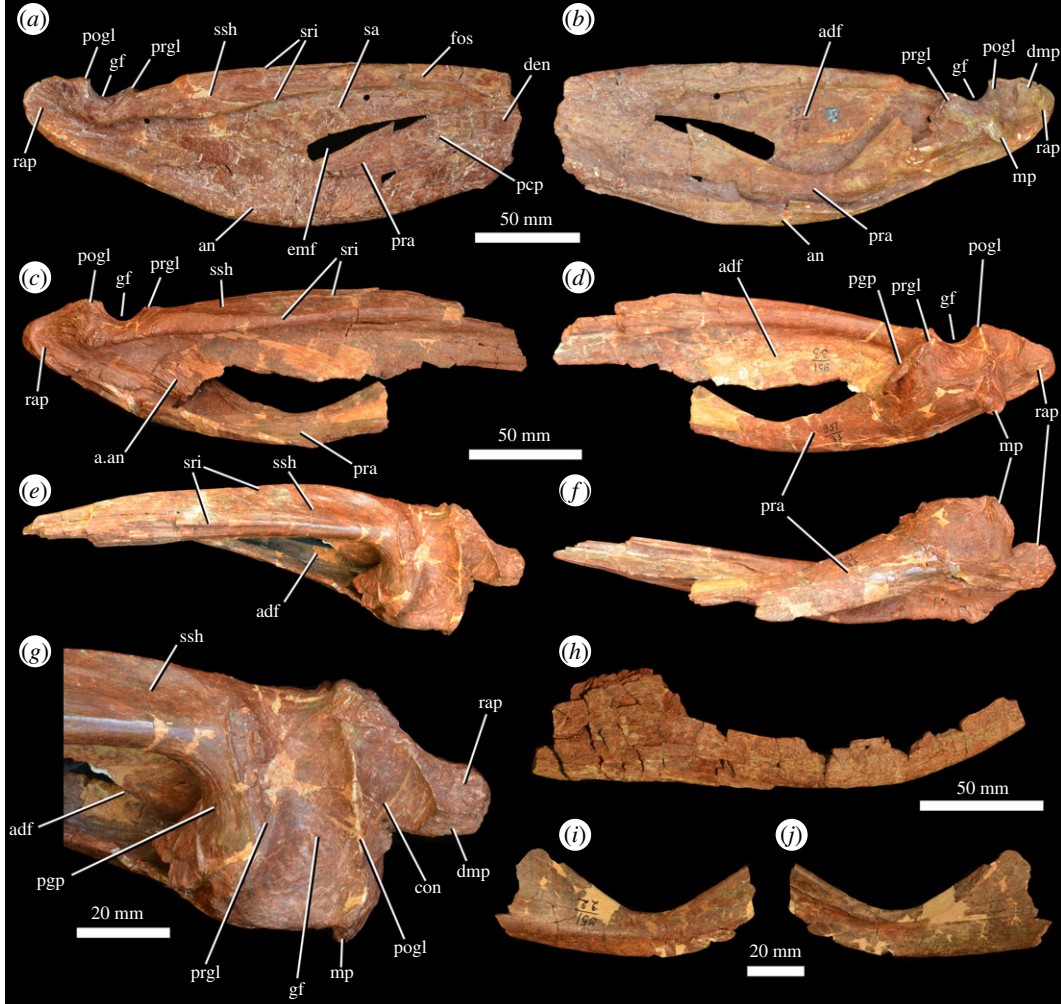

**Figure 15.** Post-dentary mandibular material of *G. prima* (='*V. triplicostata*'). Posterior right mandible (PIN 951/46-2) in lateral (*a*) and medial (*b*) views. Posterior right mandible (PIN 951/33-1) in lateral (*c*), medial (*d*), dorsal (*e*), ventral (*f*) views and close-up of the articular region in dorsal view (*g*). Partial left angular (PIN 951/30-2) in lateral view (*h*). Partial left prearticular (PIN 951/22-1) in medial (*i*) and lateral (*j*) views. a.an, articular surface for the angular; adf, adductor fossa; an, angular; con, concavity; den, dentary; dmp, dorsomedial process; emf, external mandibular fenestra; fos, fossa; gf, glenoid fossa; mp, medial process of articular; pcp, posterocentral process of the dentary; pgp, pre-glenoid process of surangular; pogl, post-glenoid lip; pra, prearticular; prgl, pre-glenoid lip; rap, retroarticular process; sa, surangular; sri, ridges on the surangular; ssh, surangular shelf.

for a short distance. Dorsal to this central process, the contact of the dentary with the surangular is unclear. However, in the holotype of *G. prima*, there is a long posterodorsal process that fits into an elongate furrow on the surangular (PIN 2394/5; [10]).

## 5.18. Splenial

A nearly complete right splenial is in articulation with the medial surface of the dentary in PIN 951/46-1 (figure 13*b*), although it is missing its anteriormost and posterior portions. A partial left splenial is preserved in PIN 951/30-1 (figure 14*b*), also in articulation with a dentary. A very poorly preserved and heavily fragmented splenial is preserved in articulation with the dentary of PIN 951/69-1. The splenial is transversely compressed and sheet-like. Anteriorly, it covers most of the medial surface of the dentary, except for a small ventral portion. More posteriorly, the splenial expands in dorsoventral height to cover the entire medial surface of the dentary and forms the ventral margin of the hemimandible posterior to the tooth row. At this point, a small portion of the splenial is visible in lateral view (figure 13*a*), as in the holotype of *G. prima* (PIN 2394/5; [10]). The medial surface is gently concave dorsoventrally at its posterior end. More anteriorly, there is a concave ventral portion

where the splenial covers the Meckelian canal, and dorsal to this region the surface of the splenial is very gently convex. The posterior contact of the splenial with the surangular and angular are uncertain because the splenial appears to have broken away on the preserved posterior half of PIN 951/46-1. No foramina appear to be present on the medial surface of the splenial; however, in PIN 951/46-1, there are two crater-like openings with raised margins positioned immediately below tooth positions 10 and 11 (figure 13b: cfor). These are interpreted as probable pathologies.

## 5.19. Surangular

There are multiple relatively complete examples of the surangular (PIN 951/30-3, PIN 951/33-1, 2, PIN 951/46-2, PIN 951/69-6; figure 15a–g). The surangular forms the dorsal margin of the posterior half of the hemimandible and the majority of the dorsal margin of the external mandibular fenestra. Anteriorly, its contacts with the dentary are unclear. Ventrally, it forms an elongate arched contact with the angular, and posteriorly, it has an extensive contact with the articular. Two ridges are present dorsally on the surangular (figure 15a,c,e). The more ventral of these ridges is well developed and overhangs the lateral surface of the surangular along much of its length, extending posteriorly to just behind the glenoid. Medial to this ridge, the dorsal portion of the surangular is transversely expanded. The second ridge is subtler and is dorsally positioned, beginning close to the anterior end of the surangular and extending along the transversely expanded dorsal portion of the bone. Above the external mandibular fenestra, an anteriorly opening fossa is present between these two ridges (figure 15a: fos). The surangular has a medial process posteriorly, the pre-glenoid process (figure 15d,g: pgp), which forms the anterior wall of the glenoid, articulating with the anterior margin of the articular bone. The glenoid extends onto the posterior part of the surangular.

At its posterior end, the surangular forms the lateral surface of the retroarticular process. In this area, the surface of the surangular is strongly rugose (figure 15a,c). The posterior end of the surangular is rounded in lateral view. The medial surface of the surangular forms much of the medial wall of the adductor fossa. The dorsal margin of the adductor fossa is defined by a thick longitudinal ridge. No foramina appear to be present on the lateral surface of the surangular in most examples of the bone. However, there are two possible foramina on the lateral surface of PIN 951/69-6, one positioned ventral to the glenoid and one anteroventral to the glenoid. In PIN 951/69-6 and the right surangular of PIN 951/33-1, the articular surface for the angular is exposed (figure 15c: a.an). The angular would have extensively overlapped the surangular laterally, and this articular surface is strongly striated with elongate ridges.

## 5.20. Angular

At least six angulars are preserved (PIN 951/16, PIN 951/30-2, PIN 951/33-3, PIN 951/46-2, PIN 951/69-2, 5; figure 15a,b,h). The angular forms the entire ventral margin of the hemimandible posterior to the dentary and splenial, and this margin is strongly convex in lateral view. Anteriorly, it forms the entire ventral border of the external mandibular fenestra. The ventral margin of the angular is transversely thickened and the lateral surface of the bone is covered with low rugosities. Anteriorly, the angular is laterally overlapped by the dentary, and medially, it would have been overlapped by the splenial. However, the splenial is not preserved in articulation in any of the specimens. Medially, the angular also forms a contact with the prearticular. At its posterior end, the angular contacts the articular.

## 5.21. Prearticular

A complete right prearticular is preserved in PIN 951/46-2 (figure 15b), part of the right prearticular is present in PIN 951/33-1 (figure 15c,d), and an isolated left prearticular is also preserved (PIN 951/22-1; figure 15i,j). The prearticular has a relatively straight ventral margin and a strongly concave dorsal margin, being contracted in its mid-portion and strongly dorsoventrally expanded at both ends. The medial surface of the bone is convex and the lateral surface is strongly concave where it formed the medial wall of the adductor fossa. The ventral portion of the medial surface bears an extensive facet for articulation with the angular. This surface is gently striated. At its posterior end, the prearticular contacts the articular medial to the glenoid, and also probably formed a contact with the pre-glenoid process of the surangular.

## 5.22. Articular

The articular is preserved in multiple specimens including PIN 951/46-2 (figure 15a,b), PIN 951/33-1, 2 (figure 15c–g), PIN 951/30-3 and PIN 951/69-6. The articular sits within a cup supported by the pre-glenoid process of the surangular anteriorly (figure 15g: pgp), the prearticular medially and ventrally, and the main body of the surangular laterally. The articular forms the majority of the glenoid (figure 15g: gf), which has a saddle-shaped surface. The pre-glenoid lip (figure 15g: prgl) is formed partly by the articular medially, but primarily by the surangular. The post-glenoid lip (figure 15g: pogl) is formed entirely by the articular. Posterior to the post-glenoid lip is a concave surface (figure 15g: con), which is continuous with a distinctly developed medial process (figure 15b,d,g: mp). This process extends a short distance medial to the glenoid. The articular forms the majority of the retroarticular process, from which a dorsomedial process extends (figure 15b,g: dmp), as also occurs in the holotype of *G. prima* (PIN 2394/5; [10]) and in *G. madiba* (NM QR 3051).

# 6. Discussion

## 6.1. Taxonomic status of '*Vjushkovia triplicostata*'

The cranial material of '*V. triplicostata*' described here is of similar size and is almost identical in morphology to the type specimen of *G. prima* [10]. As discussed previously [10], the two taxa share a substantial number of anatomical features not present in other erythrosuchids. A number of minor differences between the two taxa have been noted in the past (e.g. [10,15]), and some additional differences have been noted here. The majority of these are best explained by differences in preservation, particularly the transverse compression of the holotype skull of *G. prima* (PIN 2394/5). However, two proposed differences are worthy of further discussion.

As described above, the pterygoid and palatine of '*V. triplicostata*' both bear a small number of palatal teeth. These teeth are very small and poorly preserved. By contrast, palatal teeth have not been identified in the holotype of *G. prima* [10]. Although at face value, this appears to represent a substantial difference between the two taxa, the palate of the *G. prima* holotype is incompletely exposed and surface bone preservation is not perfect. Therefore, very small palatal teeth might be difficult to identify even if they were present. Moreover, as discussed above, there seems to be some variation in the palatal teeth among the cranial remains of '*V. triplicostata*' (potentially within the same individual), and intraspecific variation in the presence of palatal teeth occurs in some modern reptiles [10]. Given the overwhelmingly strong similarities between the material of *G. prima* and '*V. triplicostata*', we interpret the difference in the presence or absence of palatal teeth, if real, as intraspecific variation.

A second difference between *G. prima* and '*V. triplicostata*' that has been discussed previously is tooth implantation [10]. In *G. prima*, implantation is entirely thecodont, with teeth set in deep sockets but not fused to the surrounding bone. In '*V. triplicostata*', implantation is largely ankylothecodont, with the bases of the teeth fusing to the surrounding bones. However, as noted above, there is some variation in the degree of ankylothecodonty in the material of '*V. triplicostata*', even in the same jaw, and some tooth positions have a more thecodont implantation pattern (e.g. the ninth maxillary tooth of PIN 951/34). Ezcurra *et al.* [10] discussed similar variation in the development of ankylothecodont dentition in other early archosauromorphs and concluded that there may be some intraspecific variation in this character, and that is also probably the case here.

In summary, we view the differences in morphology between *G. prima* and '*V. triplicostata*' as minor, and explicable as a combination of intraspecific and taphonomic variation. Thus, we agree with some previous authors [10,15] in considering '*V. triplicostata*' [16] as a junior subjective synonym of *G. prima* [14]. As a result, the lectotype and paralectotypes of '*V. triplicostata*' are here referred to *G. prima* because of the presence of the following autapomorphies of the species (*sensu* [10]): nasal with an anteroposteriorly long descending process that forms an extensive longitudinal suture with the maxilla; prefrontal strongly flared laterally in dorsal view; skull roof with a long, median longitudinal fossa on its dorsal surface; and basioccipital with a median tuberosity on its ventral surface. The only other species previously classified in the genus *Vjushkovia* is *Vjushkovia sinensis* [57] from the Middle Triassic of China. This is not congeneric with '*V. triplicostata*' and is currently classified as *Youngosuchus sinensis* [20] and has been recovered as a pseudosuchian archosaur in recent phylogenetic analyses ([7] and iterative modifications of this analysis).

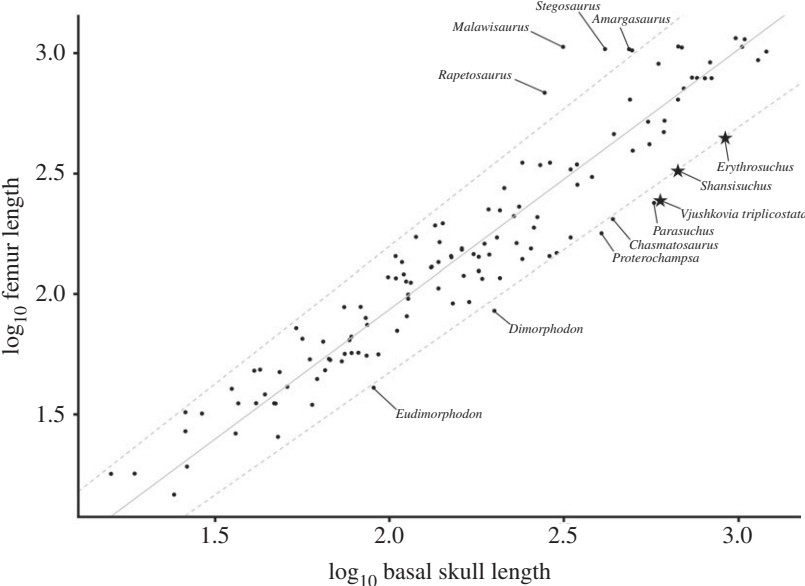

**Figure 16.** Standard major axis regression of basal skull length compared with femur length for 130 species of fossil and living tetrapods. Dashed lines indicate 95% CIs. Star symbols represent erythrosuchids.

Data from specimens of 'V. triplicostata' presented here expand our understanding of the cranial anatomy of G. prima by providing information on parts of the skull that are inaccessible in the articulated skull of the type specimen of the latter. This grants a window into individually variable structures within the species, and indeed in erythrosuchids more generally. Given the excellent holotype material and the now referred material previously attributed to 'V. triplicostata', G. prima is currently probably the best-known erythrosuchid.

## 6.2. Relative head size in erythrosuchids

Across the full dataset of 130 tetrapods, standard major axis regression recovers a strong positive correlation between basal skull length and femur length ($r^2 = 0.89$) (figure 16). The erythrosuchids 'V. triplicostata', E. africanus and S. shansisuchus, the proterosuchid 'Chasmatosaurus' yuani, the pterosaurs Dimorphodon macronyx and Eudimorphodon macronyx, the proterochampsian Proterochampsa barrionuevoi, and the phytosaur Parasuchus hislopi all have skulls that are disproportionately elongate relative to the femur. Several taxa in the dataset have disproportionately short skulls, including the sauropods Rapetosaurus krausei, Malawisaurus dixeyi and Amargasaurus cazaui, and the stegosaur Stegosaurus stenops.

Phylogenetic generalized least-squares regression of the reduced dataset of 41 Early Triassic to earliest Jurassic archosauromorph species also recovered a strongly significant positive correlation between basal skull length and femur length ($r^2 = 0.84$; $p \leq 0.0001$). All five of the phylogenetic signal statistics calculated by the phyloSignal function (Blomberg's $K$ and $K^*$, Abouheif's Cmean, Moran's $I$ and Pagel's Lambda) show significant phylogenetic signal in the skull–femur ratio data ($p = 0.01$ to $p \leq 0.0001$), and the phylogenetic correlogram suggests that this signal is lost at phylogenetic distances greater than 16 Myr. Two clear phylogenetic associations occur within the data (figure 17): proterosuchids (Proterosuchus fergusi, 'Chasmatosaurus' yuani) and erythrosuchids ('V. triplicostata', E. africanus, S. shansisuchus) form a phylogenetic cluster of taxa with proportionately large skulls, whereas the early dinosaurs Buriolestes schultzi, Eoraptor lunensis and Herrerasaurus ischigualastensis form a phylogenetic cluster of taxa with proportionately small skulls.

These results provide support for the generalization that the heads of erythrosuchids are disproportionately large relative to other groups of archosauromorphs, but also suggest that this feature is not unique to erythrosuchids but characterizes a broader grouping of early archosauriforms, including proterosuchids. The enlarged heads of proterosuchids and erythrosuchids were acquired coincident with increases in overall body size, and anatomical features related to carnivorous

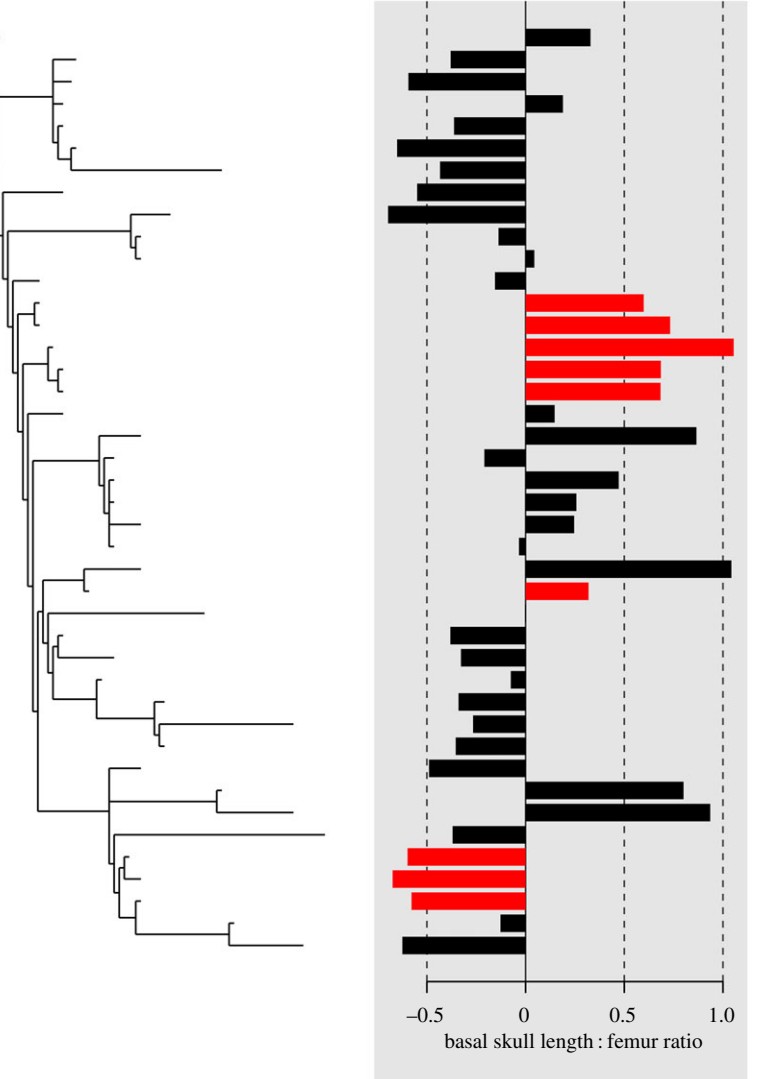

**Figure 17.** Phylogenetic association of the continuous character basal skull length : femur length ratio. Ladderized time calibrated phylogenetic supertree (left); local Moran's index values for each terminal, in which red bars indicate significant local indicators of phylogenetic association that denote species with similar neighbours or species with different neighbours (centre); and supertree terminal labels (right).

adaptations such as serrated teeth [6,7]. The evolution of large heads in erythrosuchids and other early archosauriforms might therefore be linked to these groups invading hypercarnivorous niches following the Permo-Triassic mass extinction, although further exploration of this hypothesis using functional morphological approaches is warranted.

Data accessibility. All underlying data are provided as electronic supplementary material.

Authors' contributions. R.J.B., A.G.S., E.M.D., M.D.E., B.P.H., S.C.R.M., L.E.M., T.J.R. and D.J.G. collected and interpreted anatomical data and photographs from fossil specimens. R.J.B., A.G.S., M.D.E. and D.J.G. made taxonomic interpretations of fossil data. R.J.B. drafted the manuscript and the majority of the figures. E.M.D. drafted figure 1. M.D.E. and R.J.B. carried out statistical analyses. All authors contributed to the writing of the manuscript. All authors read the final version of the manuscript, gave approval for submission and agree to be held accountable for the work performed therein.

Competing interests. We declare we have no competing interests.

Funding. This research was supported by an International Exchange grant co-funded by the Royal Society (grant no. IEC\R2\170064 to R.J.B.) and the Russian Foundation for Basic Research (RFBR; grant no. 17-54-10013 to A.G.S.). A.G.S. was also funded by the Russian Foundation for Basic Research through the research project no. 17-04-00410, and by a subsidy of the Russian Government to support the Program of 'Competitive Growth of Kazan Federal University among World's Leading Academic Centers'.

Acknowledgements. D.J.G. thanks Mike Benton for supervising his NERC scholarship-funded PhD research on erythrosuchids. We thank the editor, Julia Desojo, Spencer Lucas and an anonymous reviewer for helpful comments on an earlier version of this manuscript.

# Appendix A. List of cranial and mandibular material referred here to *Garjainia prima*

Specimens are paralectotypes of '*V. triplicostata*' unless otherwise noted

PIN 951/9      left jugal
PIN 951/13-1  fragment of left surangular
PIN 951/13-2  fragment of left surangular
PIN 951/15-1  left pterygoid
PIN 951/15-2  fragment of right pterygoid
PIN 951/16     right angular
PIN 951/17-1  incomplete left quadratojugal
PIN 951/17-2  incomplete left quadratojugal
PIN 951/18-1  right palatine
PIN 951/18-2  partial left palatine
PIN 951/19-1  left nasal
PIN 951/19-2  right nasal
PIN 951/22-1  left prearticular
PIN 951/22-2  fragment of left prearticular
PIN 951/22-3  fragment of right prearticular
PIN 951/23-1  incomplete left jugal
PIN 951/23-2  incomplete right jugal in articulation with a right quadratojugal
PIN 951/30-1  left dentary with a partial splenial
PIN 951/30-2  left angular
PIN 951/30-3  posterior left hemimandible
PIN 951/32     left maxilla
PIN 951/33-1  posterior right hemimandible
PIN 951/33-2  posterior left hemimandible
PIN 951/33-3  right angular
PIN 951/34     right maxilla
PIN 951/46-1  anterior right hemimandible (dentary + splenial)
PIN 951/46-2  posterior right hemimandible
PIN 951/54-1  left dentary
PIN 951/54-2  left dentary fragment
PIN 951/54-3  right dentary
PIN 951/55     left maxilla
PIN 951/56     fragment of right parietal?
PIN 951/57-1  left quadrate
PIN 951/57-2  left quadrate
PIN 951/57-3  left quadrate
PIN 951/59     skull roof and braincase (lectotype of '*Vjushkovia triplicostata*')
PIN 951/60     incomplete skull roof and braincase
PIN 951/63     left and right premaxillae in articulation
PIN 951/69-1  right dentary and splenial (referred specimen)
PIN 951/69-2  right angular (referred specimen)
PIN 951/69-3  fragment of the left dentary (referred specimen)
PIN 951/69-4  fragment of the left prearticular (referred specimen)
PIN 951/69-5  left angular (referred specimen)
PIN 951/69-6  posterior left hemimandible (referred specimen)
PIN 951/116   fragment of left premaxilla
PIN 951/117   incomplete right jugal
PIN 951/118   incomplete left squamosal
PIN 951/119   incomplete left squamosal
PIN 951/120   incomplete right squamosal

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
