## [Reviewer comments · Royal Society Open Science]

Review History

RSOS-191289.R0 (Original submission)

Review form: Reviewer 1 (Spencer G. Lucas)

Is the manuscript scientifically sound in its present form?

No

Are the interpretations and conclusions justified by the results?

No

Is the language acceptable?

Yes

Do you have any ethical concerns with this paper?

No

Have you any concerns about statistical analyses in this paper?

No

Recommendation?

Accept with minor revision (please list in comments)

Comments to the Author(s)

This article describes specimens of an erythrosuchid that merit more complete description and illustration. It reinforces previously advocated taxonomy, and analyzes the unusually large head size of this erythrosuchid. Unfortunately, that analysis leads to an off the cuff, unsupported conclusion about the biology of the erythrosuchid. Nevertheless, if that and a few other blemishes are fixed, this paper should be published.

Spencer G. Lucas

Specific suggestions are keyed to numbers on the manuscript (see the pdf (Appendix A)).

1. This is open to question, especially in light of the work of Gastaldo et al. (see also my article in Earth-Science Reviews in 2017). So, the authors should at least add something like this: "(but see Gastaldo et al., 2015; Lucas, 2017)."
2. Really – what ammonites from these wholly nonmarine rocks? There should be a more accurate explanation of the age of this assemblage, particularly based on the vertebrates, themselves, which I and others have published more than once.
3. There should be a synonymy here, as a taxonomic judgment is rendered.
4. So, why is this taxon called Vjushkovia in the title and text, if it is Garjainia. Those uses should be corrected to call it Garjainia.
5. ??? Why does having a large head make you an apex predator, especially when so many large predators do not have disproportionately large heads? It must have to do with the manner of killing and feeding, but is not an indicator of apex predator. So, this needs a more biologically sound/plausible conclusion.

Review form: Reviewer 2 (Felipe Lima Pinheiro)**Is the manuscript scientifically sound in its present form?**

Yes

Are the interpretations and conclusions justified by the results?

Yes

Is the language acceptable?

Yes

Do you have any ethical concerns with this paper?

No

Have you any concerns about statistical analyses in this paper?

No

Recommendation?

Accept with minor revision (please list in comments)

Comments to the Author(s)

This interesting MS assess the taxonomy of the erythrosuchid *Garjainia prima* through a detailed evaluation of a series of specimens previously referred as "*Vjushkovia triplicostata*". As a consequence of the thorough anatomical descriptions, the knowledge about *G. prima* is substantially enhanced, making this taxon the best known erythrosuchid thus far. The main statement (that is, "*V. triplicostata*" is a junior synonym of *G. prima*) is well justified by the results, and I strongly agree with author's interpretations. The paper is fluid and forms a logical integrity. It is written in a clear language and do not appear to contain significative grammatical or stylistic issues (I'm not a native English speaker myself and probably not the best person to evaluate this). The provided statistical analysis is sound and followed by accessible raw data. As such, I strongly recommend publication following the consideration of some minor issues I list below:

- The main conclusion of the MS is that "*Vjushkovia triplicostata*" is a junior synonym of *Garjainia prima*. As such, "*V. triplicostata*" should be written using quotation marks throughout the text, including the title and the Systematic Palaeontology section. In this latter, the specimens should be referred as *Garjainia prima*, and I suggest the authors to include a synonym list to include "*Vjushkovia triplicostata*".
- I suggest the authors to include a more detailed geological description of the outcrops where the holotype of *G. prima* and the referred specimens described by them were found, including a brief report of the depositional environments.
- Line 251. It is not clear if this foramen is present in the right counterpart. If this is the case, please indicate in the figure.
- Line 257. Only the anterior portion of the maxilla is downturned, as the alveolar margin below tooth positions 4-5 remain horizontal.
- Line 507. Do the authors mean 'anteroposteriorly thickened'?
- Line 605. Actually, the posterior end of the dentary is preserved in specimen PIN 951/46-2.
- Lines 607-609. It is not possible to discern dentary outlines through photographs alone. Line drawings would be useful here (see below).
- Line 639. Change to 'the more ventral of these ridges is well developed (...)'

Figures:

I believe that interpretative line drawings of complex elements would substantially improve this work, as some bones are really hard to delimitate through photographs alone.

It would be also useful to provide detailed photographs of some individual teeth.

Please include scalebars for both maps depicted in Figure 1. I also suggest the authors to indicate the horizons where *G. prima* holotype and "*Vjushkovia triplicostata*" where found in the stratigraphic column.

Please label the main jugal processes in Figure 7.

Decision letter (RSOS-191289.R0)

02-Oct-2019

Dear Dr Butler,

On behalf of the Editors, I am pleased to inform you that your Manuscript RSOS-191289 entitled "Cranial anatomy and taxonomy of the erythrosuchid archosauriform *Vjushkovia triplicostata* Huene, 1960 from the Early Triassic of European Russia" has been accepted for publication in Royal Society Open Science subject to minor revision in accordance with the referee suggestions. Please find the referees' comments at the end of this email.

The reviewers and handling editors have recommended publication, but also suggest some minor revisions to your manuscript. Therefore, I invite you to respond to the comments and revise your manuscript.

- Ethics statement

- Data accessibility

If you wish to submit your supporting data or code to Dryad (<http://datadryad.org/>), or modify your current submission to dryad, please use the following link:
<http://datadryad.org/submit?journalID=RSOS&manu=RSOS-191289>

- Competing interests

- Authors' contributions

AB carried out the molecular lab work, participated in data analysis, carried out sequence alignments, participated in the design of the study and drafted the manuscript; CD carried out

the statistical analyses; EF collected field data; GH conceived of the study, designed the study, coordinated the study and helped draft the manuscript. All authors gave final approval for publication.

- Acknowledgements

- Funding statement

Because the schedule for publication is very tight, it is a condition of publication that you submit the revised version of your manuscript before 11-Oct-2019. Please note that the revision deadline will expire at 00.00am on this date. If you do not think you will be able to meet this date please let me know immediately.

- 1) A text file of the manuscript (tex, txt, rtf, docx or doc), references, tables (including captions) and figure captions. Do not upload a PDF as your "Main Document";
- 2) A separate electronic file of each figure (EPS or print-quality PDF preferred (either format should be produced directly from original creation package), or original software format);
- 3) Included a 100 word media summary of your paper when requested at submission. Please ensure you have entered correct contact details (email, institution and telephone) in your user account;
- 4) Included the raw data to support the claims made in your paper. You can either include your data as electronic supplementary material or upload to a repository and include the relevant doi within your manuscript. Make sure it is clear in your data accessibility statement how the data can be accessed;

5) All supplementary materials accompanying an accepted article will be treated as in their final form. Note that the Royal Society will neither edit nor typeset supplementary material and it will be hosted as provided. Please ensure that the supplementary material includes the paper details where possible (authors, article title, journal name).

Kind regards,
Andrew Dunn
Royal Society Open Science
openscience@royalsociety.org

on behalf of Dr Julia Brenda Desojo (Associate Editor) and Jon Blundy (Subject Editor)
openscience@royalsociety.org

Reviewer comments to Author:

Reviewer: 1
Comments to the Author(s)

This article describes specimens of an erythrosuchid that merit more complete description and illustration. It reinforces previously advocated taxonomy, and analyzes the unusually large head size of this erythrosuchid. Unfortunately, that analysis leads to an off the cuff, unsupported conclusion about the biology of the erythrosuchid. Nevertheless, if that and a few other blemishes are fixed, this paper should be published.

Spencer G. Lucas

Specific suggestions are keyed to numbers on the manuscript (see the pdf).

1. This is open to question, especially in light of the work of Gastaldo et al. (see also my article in

Earth-Science Reviews in 2017). So, the authors should at least add something like this: “(but see Gastaldo et al., 2015; Lucas, 2017).”

2. Really – what ammonites from these wholly nonmarine rocks? There should be a more accurate explanation of the age of this assemblage, particularly based on the vertebrates, themselves, which I and others have published more than once.
3. There should be a synonymy here, as a taxonomic judgment is rendered.
4. So, why is this taxon called *Vjushkovia* in the title and text, if it is *Garjainia*. Those uses should be corrected to call it *Garjainia*.
5. ??? Why does having a large head make you an apex predator, especially when so many large predators do not have disproportionately large heads? It must have to do with the manner of killing and feeding, but is not an indicator of apex predator. So, this needs a more biologically sound/plausible conclusion.

Reviewer: 2

Comments to the Author(s)

This interesting MS assess the taxonomy of the erythrosuchid *Garjainia prima* through a detailed evaluation of a series of specimens previously referred as “*Vjushkovia triplicostata*”. As a consequence of the thorough anatomical descriptions, the knowledge about *G. prima* is substantially enhanced, making this taxon the best known erythrosuchid thus far. The main statement (that is, “*V. triplicostata*” is a junior synonym of *G. prima*) is well justified by the results, and I strongly agree with author’s interpretations. The paper is fluid and forms a logical integrity. It is written in a clear language and do not appear to contain significative grammatical or stylistic issues (I’m not a native English speaker myself and probably not the best person to evaluate this). The provided statistical analysis is sound and followed by accessible raw data. As such, I strongly recommend publication following the consideration of some minor issues I list below:

- The main conclusion of the MS is that “*Vjushkovia triplicostata*” is a junior synonym of *Garjainia prima*. As such, “*V. triplicostata*” should be written using quotation marks throughout the text, including the title and the Systematic Palaeontology section. In this latter, the specimens should be referred as *Garjainia prima*, and I suggest the authors to include a synonym list to include “*Vjushkovia triplicostata*”.

- I suggest the authors to include a more detailed geological description of the outcrops where the holotype of *G. prima* and the referred specimens described by them were found, including a brief report of the depositional environments.

- Line 251. It is not clear if this foramen is present in the right counterpart. If this is the case, please indicate in the figure.

- Line 257. Only the anterior portion of the maxilla is downturned, as the alveolar margin below tooth positions 4-5 remain horizontal.

- Line 507. Do the authors mean ‘anteroposteriorly thickened’?

- Line 605. Actually, the posterior end of the dentary is preserved in specimen PIN 951/46-2.

- Lines 607-609. It is not possible to discern dentary outlines through photographs alone. Line drawings would be useful here (see below).

- Line 639. Change to ‘the more ventral of these ridges is well developed (...)’.

Figures:

I believe that interpretative line drawings of complex elements would substantially improve this work, as some bones are really hard to delimitate through photographs alone.

It would be also useful to provide detailed photographs of some individual teeth.

Please include scalebars for both maps depicted in Figure 1. I also suggest the authors to indicate the horizons where *G. prima* holotype and "*Vjushkovia triplicostata*" were found in the stratigraphic column.

Please label the main jugal processes in Figure 7.

Author's Response to Decision Letter for (RSOS-191289.R0)

See Appendix B.

Decision letter (RSOS-191289.R1)

23-Oct-2019

Dear Dr Butler,

I am pleased to inform you that your manuscript entitled "Cranial anatomy and taxonomy of the erythrosuchid archosauriform '*Vjushkovia triplicostata*' Huene, 1960 from the Early Triassic of European Russia" is now accepted for publication in Royal Society Open Science.

Kind regards,

on behalf of Dr Julia Brenda Desojo (Associate Editor) and Jon Blundy (Subject Editor)
openscience@royalsociety.org

1
2
3 1 **Cranial anatomy and taxonomy of the erythrosuchid archosauriform *Vjushkovia***
4
5 ***triplicostata* Huene, 1960 from the Early Triassic of European Russia**
6
7
8
9

10 4 Richard J. Butler¹, Andrey G. Sennikov^{2,3}, Emma M. Dunne¹, Martin D. Ezcurra^{1,4}, Brandon
11 Hedrick^{5,6}, Susannah C. R. Maidment^{1,7}, Luke E. Meade¹, Thomas J. Raven^{7,8}, David J. Gower⁹

12
13 5
14
15 6 ¹*School of Geography, Earth and Environmental Sciences, University of Birmingham,*
16
17 *Edgbaston, Birmingham B15 2TT, UK*
18 7

19
20 8 ²*Borissiak Paleontological Institute RAS, Profsoyuznaya 123, Moscow 117647, Russia*

21
22 9 ³*Kazan Federal University, Institute of Geology and Petroleum Technologies, Kremlyovskaya*
23
24 *str. 4, Kazan 420008, Russia*
25 10

26
27 11 ⁴*Sección Paleontología de Vertebrados, CONICET – Museo Argentino de Ciencias Naturales*
28
29 *‘Bernardino Rivadavia’, Ángel Gallardo 470 (C1405DJR), Buenos Aires, Argentina*
30 12

31
32 13 ⁵*Department of Earth Sciences, University of Oxford, Oxford, OX1 3AN, UK*

33
34 14 ⁶*Department of Cell Biology and Anatomy, School of Medicine, Louisiana State University*
35
36 *Health Sciences Center, New Orleans LA 70112, USA*
37 15

38
39 16 ⁷*Department of Earth Sciences, The Natural History Museum, London SW7 5BD, UK*

40
41 17 ⁸*School of Environment and Technology, University of Brighton, Brighton BN1 4JG, UK*

42
43 18 ⁹*Department of Life Sciences, The Natural History Museum, London SW7 5BD, UK*
44
45 19

46
47 20

48
49 21

50
51 22

52
53 23

54
55 24

1
2
3 49 **Introduction**
4

5 50 Following the devastating Permo-Triassic mass extinction (PTME), terrestrial Early Triassic
6
7
8 51 ecosystems witnessed the beginnings of a major evolutionary radiation of archosauromorph
9
10
11 52 reptiles (Ezcurra & Butler 2018). Archosauromorpha is a highly diverse clade that includes
12
13 53 dinosaurs, birds, pterosaurs and crocodylians, and dominated vertebrate niches in terrestrial
14
15 54 ecosystems throughout much of the Mesozoic (Nesbitt 2011; Ezcurra et al. 2013; Ezcurra
16
17
18 55 2016). One of the earliest archosauromorph clades to diversify following the PTME was the
19
20
21 56 Erythrosuchidae, a group of medium to large-bodied apex predators comprising
22
23 57 approximately seven species, known from South Africa, China, India, and Russia (Parrish
24
25 58 1992; Gower 2003; Ezcurra et al. 2013, 2019; Gower et al. 2014; Butler et al. 2019, in press).
26

27
28 59 The earliest definitive erythrosuchids are known from the latest Early Triassic, represented
29
30 60 by the genus *Garjainia* Ochev, 1958, including the species *Garjainia prima* Ochev, 1958 from
31
32 61 Russia, and *Garjainia madiba* Gower, Hancox, Botha-Brink, Sennikov & Butler, 2014 from
33
34 62 South Africa (Ochev 1958; Gower & Sennikov 2000; Gower et al. 2014; Ezcurra et al. 2019).
35
36

37 63 The type species of *Garjainia* (by original monotypy), *Garjainia prima* Ochev, 1958, is
38
39 64 based on a well-preserved partial skeleton comprising a nearly complete skull and
40
41
42 65 fragmentary postcranium, from the Kzyl-Sai II locality, approximately 70 km southeast of the
43
44 66 city of Orenburg, in Orenburg Province, Russia (Fig. 1). Another Russian erythrosuchid genus
45
46 67 and species, *Vjushkovia triplicostata* Huene, 1960, was erected based on well-preserved
47
48
49 68 skeletal remains representing multiple individuals collected at the Rassypnaya locality, west
50
51
52 69 of Orenburg (Figs 1, 2), about 150 km distant from Kzyl-Sai II. The fossils of these two
53
54 70 erythrosuchid taxa were both collected from similar stratigraphic horizons, within the
55
56
57 71 Petropavlovskaya Svita of the Yarengian Supergorizont (Shishkin et al. 2000), and there have
58
59 72 been repeated suggestions that they are synonymous at the generic or even specific level.
60

1
2
3 121 scientists being assisted by local school children. The fossil bones of *Vjushkovia triplicostata*
4
5
6 122 were preserved in claystone and were fragile, proving difficult to excavate and prepare. An
7
8 123 unexpected storm during the excavation covered the locality and surrounding steppe with
9
10
11 124 more than a metre of snow, meaning that fossils had to be transported to the railway
12
13 125 station using horses and carts (Fig. 2A). In 1954, B. P. Vyushkov organised a second field
14
15 126 season to continue the excavation (Fig. 2B), including the first use of a bulldozer in Russian
16
17 127 palaeontological fieldwork (Ochev 2000). Bones were collected from a minimum of six
18
19 128 individuals of *V. triplicostata*, showing a substantial size range. More recently, in 1974 M. G.
20
21 129 Minikh, a palaeontologist from Saratov State University, discovered at the same locality and
22
23 130 claystone horizon the remains of an additional large individual that he identified as *V.*
24
25 131 *triplicostata* (M. G. Minikh pers. comm. to A.G.S.). Currently, the outcrop and excavation
26
27 132 site are not exposed, being covered by landslips and vegetation (A.G.S. pers. obs.), but the
28
29 133 locality is listed as a geological monument of ^{the} Orenburg region (Fig. 2C).
30
31
32
33
34

35 134 The type specimen of *Garjainia prima* and the type material of *Vjushkovia triplicostata*
36
37 135 come from the Yarengian Supergorizont (Fig. 1), where the fauna have been subdivided into
38
39 136 two tetrapod biochrons corresponding to the Fedorovkian and Gamian gorizonts (Shishkin
40
41 137 et al. 2000). The type locality for *Garjainia prima*, Kzyl-Sai II, occurs within the lower of these
42
43 138 two biozones, the Fedorovkian. By contrast, the type locality of *Vjushkovia triplicostata*,
44
45 139 Rassypnaya, occurs with the upper biozone, the Gamian. Thus, there is a potential temporal
46
47 140 difference between the two species, with *Vjushkovia triplicostata* being stratigraphically
48
49 141 younger. Both the Fedorovkian and Gamian gorizonts are dated as latest Early Triassic (late
50
51 142 Olenekian) in age, based on ammonite and miospore biostratigraphy (Shishkin et al. 2000).
52
53
54
55
56

57 143 Associated fauna from the Rassypnaya locality include the temnospondyl *Parotosuchus*
58
59 144 *orenburgensis*, the archosauriform *Chasmatosuchus magnus* (= '*Jaikosuchus magnus*');
60

193 for each function respectively. The lipaMoran function of phylosignal, setting 999,999
 194 replicates and a significant p-value < 0.05, was used to identify phylogenetic associations
 195 within the data – i.e. groups of species with similar values – and the results were visualised
 196 using the barplot.phylo4d function.

197

198 **Systematic palaeontology**

199

200 *Diapsida* Osborn, 1903 sensu Laurin (1991)

201 *Sauria* Gauthier, 1984 sensu Gauthier, Kluge & Rowe (1988)

202 *Archosauromorpha* von Huene, 1946 sensu Dilkes (1998)

203 *Archosauriformes* Gauthier, Kluge & Rowe, 1988

204 *Erythrosuchidae* Watson, 1917 sensu Ezcurra et al. (2010)

205 *Vjushkovia* Huene, 1960

206
 207 *Vjushkovia triplicostata* Huene, 1960

208

209

210 *Lectotype*: PIN 951/59, a skull roof and braincase (Huene 1960: pl. 11, figs 1, 2; Tatarinov

211 1961: fig. 2; Gower & Sennikov 2000: fig. 8.4C).

212

213 *Paralectotypes*: Numerous isolated skull and postcranial elements, including parts of nearly

214 the entire skeleton, representing at least six individuals (see Appendix 1 for full list of

215 paralectotype cranial and mandibular material).

216

1
2
3 766 positive correlation between basal skull length and femur length ($r^2 = 0.84$; $p = < 0.0001$). All
4
5
6 767 five of the phylogenetic signal statistics calculated by the phyloSignal function (Blomberg's K
7
8 768 and K^* , Abouheif's Cmean, Moran's I, and Pagel's Lambda) show significant phylogenetic
9
10 769 signal in the skull-femur ratio data ($p = 0.01$ to $p = < 0.0001$), and the phylogenetic
11
12 770 correlogram suggests that this signal is lost at phylogenetic distances greater than 16 million
13
14 771 years. Two clear phylogenetic associations occur within the data (Fig. 17): proterosuchids
15
16 772 (*Proterosuchus fergusi*, '*Chasmatosaurus*' *yuani*) and erythrosuchids (*Vjushkovia*
17
18 773 *triplicostata*, *Erythrosuchus africanus*, *Shansisuchus shansisuchus*) form a phylogenetic
19
20 774 cluster of taxa with proportionately large skulls, whereas the early dinosaurs *Buriolestes*
21
22 775 *schultzi*, *Eoraptor lunensis*, and *Herrerasaurus ischigualastensis* form a phylogenetic cluster
23
24 776 of taxa with proportionately small skulls.

25
26
27
28
29
30 777 These results provide support for the generalisation that the heads of erythrosuchids
31
32 778 are disproportionately large relative to other groups of archosauromorphs, but also suggest
33
34 779 that this feature is not unique to erythrosuchids but characterises a broader grouping of
35
36 780 early archosauriforms, including proterosuchids. The enlarged heads of proterosuchids and
37
38 781 erythrosuchids were acquired coincident with increases in overall body size, and anatomical
39
40 782 features related to carnivorous adaptations such as serrated teeth (Ezcurra et al. 2013;
41
42 783 Ezcurra 2016). The evolution of large heads in erythrosuchids and other early
43
44 784 archosauriforms might therefore be linked to these groups invading hypercarnivorous, apex 5
45
46 785 predator niches following the Permo-Triassic mass extinction.

47
48
49
50
51
52 786

53
54 787

55
56
57 788 **Acknowledgements**

58
59 789
60

Appendix B

Reviewer: 1

Comments to the Author(s)

This article describes specimens of an erythrosuchid that merit more complete description and illustration. It reinforces previously advocated taxonomy, and analyzes the unusually large head size of this erythrosuchid. Unfortunately, that analysis leads to an off the cuff, unsupported conclusion about the biology of the erythrosuchid. Nevertheless, if that and a few other blemishes are fixed, this paper should be published.

Spencer G. Lucas

Specific suggestions are keyed to numbers on the manuscript (see the pdf).

1. This is open to question, especially in light of the work of Gastaldo et al. (see also my article in *Earth-Science Reviews* in 2017). So, the authors should at least add something like this: “(but see Gastaldo et al., 2015; Lucas, 2017).”

The Permo-Triassic mass extinction is not the focus of this paper, but we have added a citation to the impact of and recovery from the extinction and the caveat “but see Gastaldo et al., 2015; Lucas, 2017).”

2. Really—what ammonites from these wholly nonmarine rocks? There should be a more accurate explanation of the age of this assemblage, particularly based on the vertebrates, themselves, which I and others have published more than once.

We have explained that the ammonites – which provide the best evidence for age – are derived from strata that also include *Parotosuchus*, the index genus for the Yarengian.

3. There should be a synonymy here, as a taxonomic judgment is rendered.

We have added a synonymy to the Systematic Palaeontology section.

4. So, why is this taxon called *Vjushkovia* in the title and text, if it is *Garjainia*. Those uses should be corrected to call it *Garjainia*.

Reviewer 2 made a similar point – please see our response to them below.

5. ??? Why does having a large head make you an apex predator, especially when so many large predators do not have disproportionately large heads? It must have to do with the manner of killing and feeding, but is not an indicator of apex predator. So, this needs a more biologically sound/plausible conclusion.

We have removed the term “apex”, and added a caveat to this section.

Reviewer: 2

Comments to the Author(s)

This interesting MS assess the taxonomy of the erythrosuchid *Garjainia prima* through a detailed evaluation of a series of specimens previously referred as "*Vjushkovia triplicostata*". As a consequence of the thorough anatomical descriptions, the knowledge about *G. prima* is substantially enhanced, making this taxon the best known erythrosuchid thus far. The main statement (that is, "*V. triplicostata*" is a junior synonym of *G. prima*) is well justified by the results, and I strongly agree with author's interpretations. The paper is fluid and forms a logical integrity. It is written in a clear language and do not appear to contain significant grammatical or stylistic issues (I'm not a native English speaker myself and probably not the best person to evaluate this). The provided statistical analysis is sound and followed by accessible raw data. As such, I strongly recommend publication following the consideration of some minor issues I list below:

- The main conclusion of the MS is that "*Vjushkovia triplicostata*" is a junior synonym of *Garjainia prima*. As such, "*V. triplicostata*" should be written using quotation marks throughout the text, including the title and the Systematic Palaeontology section. In this latter, the specimens should be referred as *Garjainia prima*, and I suggest the authors to include a synonym list to include "*Vjushkovia triplicostata*".

We have made these suggested changes.

- I suggest the authors to include a more detailed geological description of the outcrops where the holotype of *G. prima* and the referred specimens described by them were found, including a brief report of the depositional environments.

We have not added this because we think that sufficient geological background is already included, and because the '*V. triplicostata*' site is no longer exposed to allow sedimentological work.

- Line 251. It is not clear if this foramen is present in the right counterpart. If this is the case, please indicate in the figure.

This foramen is present on both sides, and we have now indicated this on Figure 3.

- Line 257. Only the anterior portion of the maxilla is downturned, as the alveolar margin below tooth positions 4-5 remain horizontal.

We assume that the reviewer is referring to the premaxilla, and have modified the text to say "the anterior part of the premaxilla was downturned relative to the maxilla".

- Line 507. Do the authors mean 'anteroposteriorly thickened'?

Yes. We have corrected this.

- Line 605. Actually, the posterior end of the dentary is preserved in specimen PIN 951/46-2.

The reviewer is correct. We have changed this in the text.

- Lines 607-609. It is not possible to discern dentary outlines through photographs alone. Line drawings would be useful here (see below).

We have not added line drawings, because the number of figures is already very large.

- Line 639. Change to 'the more ventral of these ridges is well developed (...)'.
(...)'.

We have made this change.

Figures:

I believe that interpretative line drawings of complex elements would substantially improve this work, as some bones are really hard to delimitate through photographs alone.

We have not added line drawings, because the number of figures is already very large.

It would be also useful to provide detailed photographs of some individual teeth.

Teeth are generally poorly preserved and provide few anatomical details, which is why we have not provided detailed photographs.

Please include scalebars for both maps depicted in Figure 1. I also suggest the authors to indicate the horizons where *G. prima* holotype and "*Vjushkovia triplicostata*" where found in the stratigraphic column.

We have modified this figure according to this suggestion.

Please label the main jugal processes in Figure 7.

We have labeled these as suggested.